# Characterization of ori and parS-like functions in secondary genome replicons in *Deinococcus radiodurans*

Ganesh K Maurya[1,2] ⓘ, Hari S Misra[1,2] ⓘ

The mechanisms underlying multipartite genome maintenance and its functional significance in extraordinary radioresistance of *Deinococcus radiodurans* are not well understood. The sequences upstream to *parAB* operons in chrII (*cisII*) and MP (*cisMP*) could stabilize an otherwise, non-replicative *colE1* plasmid, in *D. radiodurans*. DnaA and cognate ParB proteins bound specifically with *cisII* and *cisMP* elements. The Δ*cisII* and Δ*cisMP* cells showed the reduced copy number of cognate replicons and radioresistance as compared with wild type. Fluorescent reporter–operator system inserted in chrI, chrII, and MP in wild type and *cisII* mutants showed the presence of all three replicons in wild-type cells. Although chrI was present in all the Δ*cisII* and Δ*cisMP* cells, nearly half of these cells had chrII and MP, respectively, and the other half had the reduced number of foci representing these replications. These results suggested that *cisII* and *cisMP* elements contain both *origin of replication* and *parS*-like functions and the secondary genome replicons (chrII and MP) are maintained independent of chrI and have roles in radioresistance of *D. radiodurans*.

## Introduction

DNA replication and segregation are the highly coordinated macromolecular events required in the growth of any organism (Badrinarayanan et al, 2015). The *origin of replication* (*oriC*) where chromosomal replication in bacteria initiates is a well-conserved region on chromosomes and comprises of the non-palindromic repeats of 9-mer as DnaA boxes and 13-mer AT-rich repeats (Ogasawara et al., 1985; Marczynski & Shapiro, 1992; Messer, 2002). In *Escherichia coli*, DnaA, a replication initiation protein, first binds to the DnaA boxes which subsequently leads to the recruitment of replication initiation complex comprising of DnaB helicase, primase, and DNA polymerase III (Tougu & Marians, 1996; McHenry, 2011). Earlier, the tripartite genome segregation comprising (i) an origin-proximal *cis*-acting (*parS*-like) DNA sequences,

(ii) *parS* binding proteins like ParB or its homologues, and (iii) the P-loop Walker ATPases such as ParA or its homologues has been reported to be the major mechanisms associated with the segregation of duplicated genome into daughter cells (Hayes & Barillà, 2006). Typically, ParB homologues first bind to the *parS*-like sequences and the interaction of ParA homologues to ParB-*parS* complex leads to either polymerization or depolymerization of ParA as required for DNA segregation (Hayes & Barillà, 2006; Gerdes et al, 2010; Reyes-Lamothe et al, 2012). Among different mechanisms of plasmid/chromosome segregation in bacteria, the diffusion-ratchet model is found to be the most acceptable (Vecchiarelli et al, 2010). Recently, it is shown that the duplicated chromosome segregates into daughter cells by the cumulative actions of different genome maintenance proteins viz. ParA-ParB, SMC (structural maintenance of chromosome)/MukB, topoisomerases, translocases, and nucleoid-associated proteins (Pióro & Jakimowicz, 2020). Furthermore, it has been observed that the initiation of DNA replication and segregation of duplicated chromosomes occur concurrently (Marczynski et al, 2019). In the majority cases where DNA replication and segregation have been studied, the *parS*-like sequences are found close to the *origin of replication* (*ori*) region. For instance, in *Bacillus subtilis* and *Caulobacter crescentus*, the primary *parS* sequences are located within 8–10-kb from *oriC* (Livny et al, 2007). Furthermore, the genome segregation proteins interact directly with DnaA in many bacteria (Murray & Errington, 2008; Marczynski et al, 2019). The organization and dynamics of nucleoid during cell growth have been studied in monopartite genome harboring rod- or crescent-shaped bacteria such as *E. coli*, *B. subtilis*, and *C. crescentus* (Esnault et al, 2007; Fisher et al, 2013; Le et al, 2013; Badrinarayanan et al, 2015; Dame & Tark-Dame, 2016) and cocci-like *Streptococcus pneumonia* (Kjos & Veening, 2017) and *Staphylococcus aureus* (Morikawa et al, 2006, 2019).

Recently, many bacteria have been reported with multipartite genome system comprising more than one chromosome and large plasmids (Misra et al, 2018). Studies on multipartite genome maintenance in bacteria are limited to *Vibrio cholerae*, *Burkholderia cenocepacia*, *Rhodobacter sphaeiroides*, and *Shinorhizobium meliloti* (Fogel & Waldor, 2005; Dubarry et al, 2006, 2019; Galardini et al, 2013). In these bacteria, the proteins of tripartite genome segregation

---

[1]Molecular Biology Division, Bhabha Atomic Research Centre, Mumbai, India   [2]Homi Bhabha National Institute, Mumbai, India

Correspondence: hsmisra@barc.gov.in
Ganesh K Maurya's present address is Zoology Section, Mahila Mahavidyalaya, Banaras Hindu University, Varanasi, India (EN:21910)

encoded on the primary chromosome are phylogenetically similar to the chromosome of monopartite genome-harboring bacteria and found to be different from secondary genome components. Because the sizes of genomic replicons are different, how their replication and segregation are synchronized before cytokinesis would have occurred is intriguing. However, the real-time monitoring of DNA replication in *V. cholerae* a multipartite genome harboring (MGH) bacteria has shown that replication initiation of different replicons is staggered to terminate roughly at the same time (Rasmussen et al, 2007; Kemter et al, 2018).

Deinococcus radiodurans, a Gram-positive nonpathogenic bacterium characterized for extraordinary radioresistance, also harbors a multipartite genome comprising chromosome I (2,648,638 bp), chromosome II (412,348 bp), megaplasmid (177,466 bp), and small plasmid (45,704 bp) (White et al, 1999). Except small plasmid, all other replicons encode ParA- and ParB-like proteins in *parAB* operons. Earlier, Minsky and colleagues demonstrated that the multipartite genome of this bacterium is packaged in the form of a doughnut-shaped toroidal structure (Levin-Zaidman et al, 2003). Recently, the nucleoid dynamics has been monitored at different stages of growth in *D. radiodurans* and observed that the highly condensed nucleoid adapts multiple configurations during cell cycle progression, and the *oriC* in chrI remains radially distributed around the centrally positioned *ter* (terminus) sites (Floc'h et al, 2019). The characterization of *ori* and *parS*-like sequences in chromosome II (chrII) and megaplasmid (MP), and their localizations in nucleoid have not been reported. Here, we report that the direct repeats located upstream of *parAB* operons in chrII (*cisII*) and MP (*cisMP*) confers both *ori* and *parS*-like functions. Furthermore, we demonstrate that DnaA encoded on chrI interacts with these *cis*-elements of secondary genome components, whereas ParB of chrII (ParB2) and megaplasmid (ParB3) shows specific interaction with cognate *cis*-elements in vitro. Interestingly, both these *cis*-elements help in the maintenance of an otherwise suicidal plasmid, in *D. radiodurans*. The deletion of *cisII* (Δ*cisII*) and *cisMP* (Δ*cisMP*) leads to reduction in copy numbers of respective replicons and the partial loss of γ-radiation resistance as compared with wild type. These *cis* mutants produce an increased frequency of cells devoid of cognate replicons. The *tetO*-TetR-GFP–based fluorescent reporter–operator system (FROS) inserted near the putative origin of replication in chrI, chrII, and MP allowed us to localize these replicons in the nucleoid. Most Δ*cisII* and Δ*cisMP* cells showed the loss of chrII and MP, respectively, whereas there was no change in the wild-type pattern of chrI in these cells. These results suggested that *cisII* and *cisMP* confer both *ori* and *parS*-like functions, and DnaA encoded on the primary chromosome seems to regulate chrII and MP replication. Nearly no change in chrI localization pattern and its copy number in both the *cis* mutants, which has affected the maintenance of cognate replicons, further concluded that primary chromosome and secondary genome replicons are maintained independently in this bacterium.

# Results

## Sequences in chrII and MP predicted as putative "*oriC*" and *parS*-like sequences

The upstream of *parAB* operons in chrII and MP contains repetitive sequences similar to the known bacterial "*oriC*" and *parS*-like cis-

elements. The region spanning between 99 and 544 bp in chrII contain 11 iterons of 18 bp with consensus sequence T(G)CCA-CAAAGTGCCA(G)CAGG and GC content of 51.6% was named as *cisII* (Fig 1A). Similarly, the region spanning 177,446–417 bp ion MP contains eight iterons of 18 bp with consensus sequence C(A)CCGCAAAGGTG(A)TCGCTA and 3 iterons of 14 bp with consensus sequence TTTTGACCCCAAAT and GC content of 47% together named as *cisMP* (Fig 1B). In addition, both *cisII* and *cisMP* elements contain non-canonical putative DnaA boxes which were different from the 13 9-mer DnaA boxes (T(A/G)TA(T)TCCACA) present in the origin of replication of chrI in *D. radiodurans* and DnaA boxes in *oriC* of *E. coli* (Messer, 2002; Maurya & Misra, 2020 Preprint). Although the DnaA boxes started just upstream to 18-bp iterons, they continued into the iterons resulting to considerable overlap between both the elements. In addition, both the *cis*-elements contain AT-rich sequences, which are the known signatures of "*ori*-"like sequences needed for replication initiation in bacteria. In a previous study, three *parS*-like regions with consensus of GTTTC(A)G(A)C(A)GT(C)GG(T)A(G)AAC were mapped at 3.5° (*seg*S1), 72.9° (*seg*S2) and 231.8° (*seg*S3) in chrI (Charaka & Misra, 2012). The predicted *parS*-like sequences in chrII and MP are found to be distinct from the conserved *seg*S sequence of chrI. Thus, we found that *cisII* and *cisMP* elements have DnaA boxes similar to "*oriC*" feature and the iterons as *parS*-like sequences. The presence of the different "*ori*" and *parS*-like signatures in chrI from that of secondary genome replicons like chrII and MP indicated a strong possibility of independent maintenance of primary chromosome and secondary genome replicons in *D. radiodurans*.

## *cisII* and *cisMP* elements help *E. coli* plasmid maintenance in *D. radiodurans*

The function of *cisII* and *cisMP* as *origin of replication* was investigated using an *E. coli* plasmid pNOKOUT (Kan^R) that was non-replicative in *D. radiodurans*. The pNOKCisII and pNOKCisMP mini replicons that contain *cisII* and *cisMP* elements, respectively, were created in pNOKOUT and their independent maintenance in *D. radiodurans* was monitored. The *D. radiodurans* cells harboring pNOKCisII or pNOKCisMP could multiply in the presence of kanamycin (6 μg/ml), which was not observed when these cells were transformed with pNOKOUT vector (Fig 2A and B). Furthermore, the independent maintenance of pNOKCisII or pNOKCisMP mini replicons was also ascertained in Δ*recA* mutant of *D. radiodurans* and by recovering the intact plasmid DNA from the recombinant cells. We observed that the Δ*recA* cells harboring pNOKCisII or pNOKCisMP replicons continued expressing kanamycin resistance like wild-type transformants of these plasmids (Fig 2C). The plasmids isolated from the wild-type transformants of *D. radiodurans* showed the typical patterns of original plasmid on an agarose gel and the release of the expected size inserts upon restriction digestion (Fig 2D). When we measured the copy number of these plasmids by qPCR, it was found to be 6.86 ± 0.71 for pNOKCisII and 12.15 ± 0.92 for pNOKCisMP. These results together suggested that pNOKCisII and pNOKCisMP mini replicons have been maintained independently in the *D. radiodurans* cell and not integrated into its genome, supporting the "*ori*" nature of *cisII* and *cisMP* elements in this bacterium. The *cis*-elements helping the non-replicative

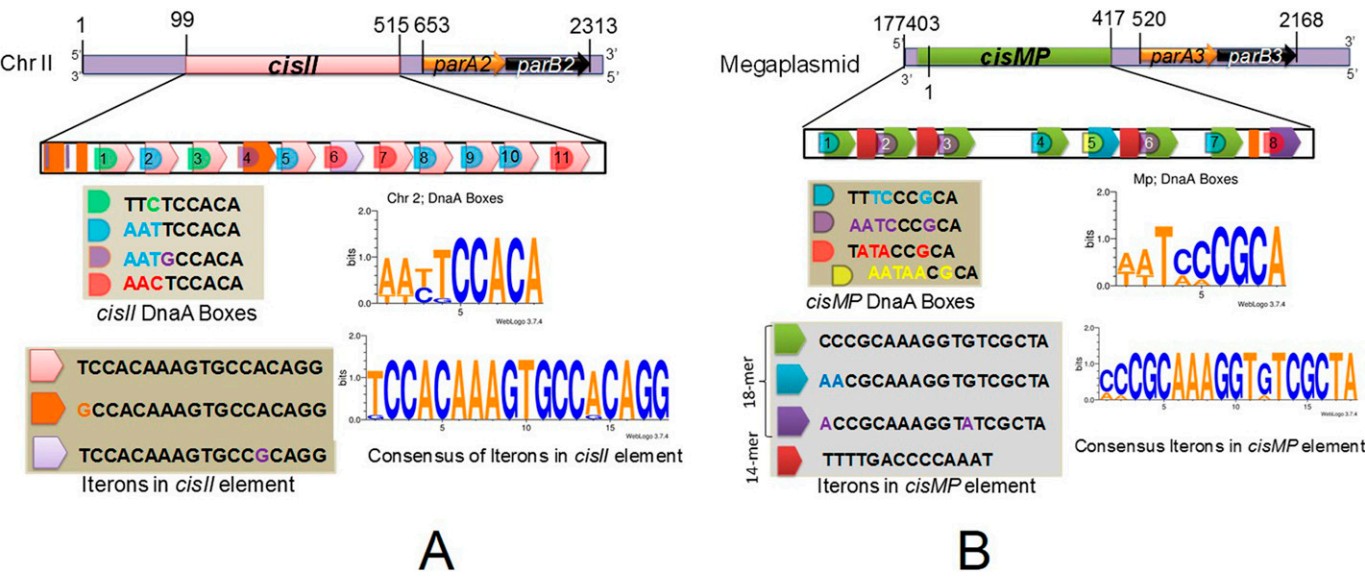

**Figure 1. Schematic presentation of direct repeats located upstream to *parAB* operon in chromosome II (*cisII*) and megaplasmid (*cisMP*).**
**(A, B)** The 99–554 bp region of chromosome II (A) and 177,446- to 417-bp region of megaplasmid (B) were analysed in silico for the structure of direct repeats and their sequence compositions of DnaA boxes and iterons.

pNOKOUT plasmid to replicate in *D. radiodurans*, the possibility of these *cis*-elements working as *parS*-like sequences and thereby helping in the segregation of these mini replicons into daughter cells was hypothesized.

### *cisII* and *cisMP* elements provide sequence-specific interaction to DnaA

The presence of the known DNA replication initiation proteins such as DnaA; RctB of *V. cholerae* (Egan & Waldor, 2003); RepC homologues as known in *Agrobacterium tumefaciens* (Pinto et al, 2011), *Brucella abortus* (Pinto et al, 2012), *Burkholderia pseudomallei* (Holden et al, 2004), and *S. meliloti* (Cervantes-Rivera et al, 2011;

Pinto et al, 2012); and RepN of *B. subtilis* (Tanaka et al, 2005) and *S. aureus* (Balson & Shaw, 1990) were searched on chrI, chrII, and MP of *D. radiodurans*. ChrII and MP do not encode for any of the known DNA replication initiators and DnaA homologue is encoded on chrI (data not given). Thus, the possibility of this DnaA interaction with *cis*-element of chrII and MP was checked. The recombinant purified DnaA (Fig S1A) showed sequence-specific interaction with both *cisII* (Fig 3A) and *cisMP* (Fig 3B) *albeit* with a different affinity. The affinity of DnaA varies as the number of repeats was reduced (Fig 3A–H and Table 1). As expected, the DnaA bound to either of the *cis*-elements remained unaffected even in the presence of an 80-fold higher molar concentration of nonspecific DNA (Fig 3). The formation of DnaA complex with *cisII* and *cisMP* containing several DnaA

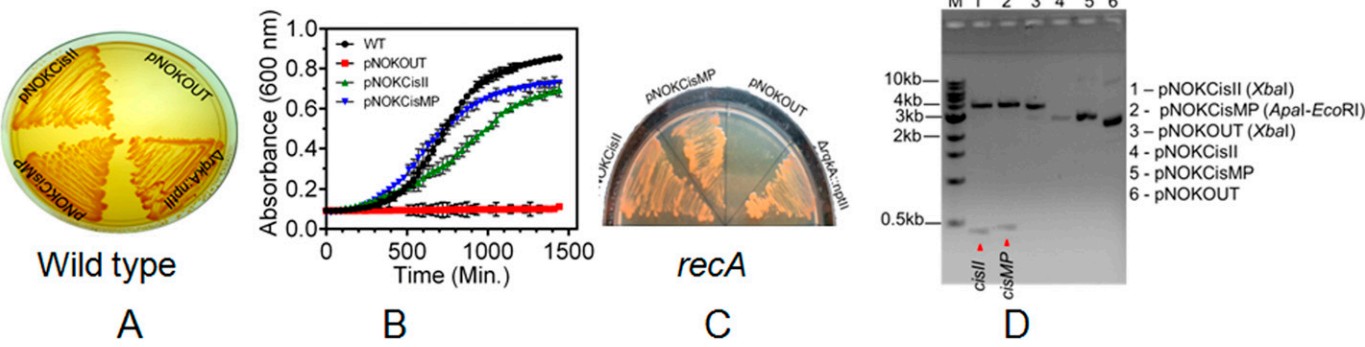

**Figure 2. Role of *cisII* and *cisMP* elements in the replication of *Escherichia coli* plasmid into *Deinococcus radiodurans*.**
**(A, C)** The *cisII* and *cisMP* elements were cloned in *E. coli* cloning vector pNOKOUT, and the resultant plasmids pNOKcisII and pNOKcisMP were transformed into wild-type (A) and *recA* mutant (C). These transformants were grown in the presence of kanamycin (6 μg/ml) and growth was monitored on TGY agar plate with *rqkA* deletion mutant (a deletion mutant made by inserting kanamycin cassettes) as described in Rajpurohit & Misra (2010) as positive control and vector as a negative control. **(B)** The growth characteristic of these transformants grown in TGY broth supplemented with antibiotic was compared with wild-type cells grown in TGY broth without antibiotics (B). **(D)** The recombinant plasmids isolated from *D. radiodurans* cells growing in the presence of antibiotic were digested with restriction enzymes, and the release of *cisII* and *cisMP* fragments was analysed on an agarose gel and compared with empty vector (D).

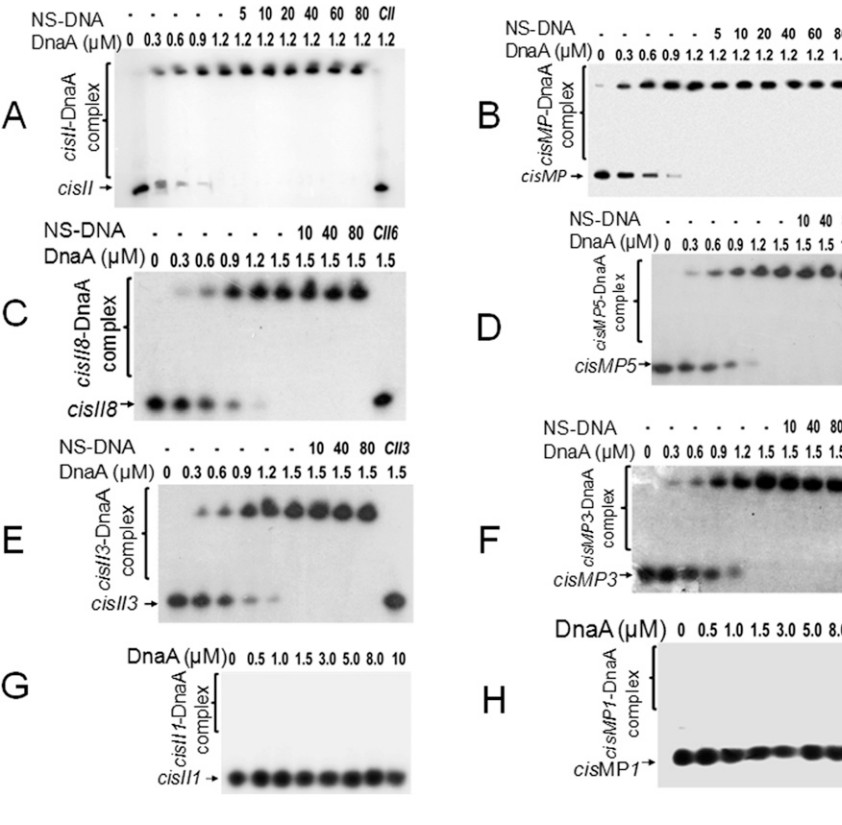

**Figure 3. DnaA interaction with *cisII* and *cisMP* and their repeats variants.**
**(A, B, C, D, E, F, G, H)** The radiolabelled full-length *cisII* containing 11 repeats (A) and *cisMP* containing eight repeats (B) as well as eight repeats (C), three repeats (E) and one repeat (G) of *cisII* (Fig S4A), and five repeats (D), three repeats (F), and single repeat (H) of *cisMP* (Fig S4B) were incubated with the increasing concentration of DnaA. A saturating concentration of the DnaA-DNA ratio was chased with increasing molar ratio of non-specific DNA (NS-DNA) and products were analysed on native PAGE. The fractions of DNA bound with proteins were quantified densitometrically and plotted as a function of protein concentration using GraphPad Prism 6. The Kd for the curve fitting of individual plots was determined and given in Table 1.

boxes suggests the "*ori*" nature of both the *cis*-elements in *D. radiodurans*.

### ParB2 and ParB3 show specific interaction with cognate *cis*-elements

Because both the *cis*-elements have iterons as a potential *parS*-like sequences and ParB homologues are known to bind *parS*-like sequences, the interactions of ParB2 and ParB3 with their cognate *cis*-elements were monitored. The radiolabelled *cisII* and their repeat variants (Fig 4A–D) as well as *cisMP* and their repeat variants (Fig 4E–H) were incubated with purified recombinant ParB2 (Fig S1B) and ParB3 proteins (Fig S1C), respectively, and the products were analysed on native PAGE. The ParB2 and ParB3 showed higher affinity with full-length *cisII* (Kd = 0.40 ± 0.007 μM) and *cisMP* (0.59 ±

0.05 μM), respectively (Fig 4A and E). The affinity of ParBs to cognate *cis*-element was nearly same up to three repeats and reduced when the number of repeats becomes less than three (Fig 4C and G and Table 2). These proteins are sparingly bound to DNA substrates with less than two repeats (Fig 4D and H). The binding of ParBs to the cognate *cis*-elements was sequence specific as it was not affected in the presence of up to 100-fold higher molar concentration of nonspecific DNA (Fig 4A–C and E–G). Furthermore, it was noticed that the full-length *cisII* and *cisMP* bound ParBs migrated under electrophoresis with peculiar patterns that are normally seen when the protein binding causes a structural change like bending in the DNA. These findings suggested that both *cisII* and *cisMP* elements have a specific region for interaction with respective ParB2 and ParB3, which might function as *parS*-like sequence in the faithful segregation of cognate replicons.

### *cisII* and *cisMP* deletion affected cognate replicons copy number and their segregation

To understand the essentiality of chrI, chrII, and MP, the possibility of deleting *cisII*, *cisMP*, and putative *ori* of chrI (*oriCI*) was explored. We could not obtain the deletion mutant of *oriCI*. However, the homozygous replacement of *cisII* (Fig S2A and B) and *cisMP* (Fig S2A and C) with an expressing cassette of *nptII* was achieved, and these cells were viable in both liquid medium and agar plates. Therefore, the effects of *cisII* and *cisMP* deletion on the stability of cognate replicons and their genome copy numbers were determined in mutant cells and compared with wild type. A significant reduction in

**Table 1. The dissociation constant (Kd) value of DnaA interaction with different repeats of *cisII* and *cisMP* elements.**

| Repeat no. | Dissociation constants (Kd) (mean ± SD, μM) | |
| --- | --- | --- |
| | *cisII* variants | *cisMP* variants |
| Full length | 0.31 ± 0.004 | 0.56 ± 0.21 |
| 8/5 repeats | 0.70 ± 0.35 | 0.89 ± 0.39 |
| 3 repeats | 0.80 ± 0.11 | 1.4 ± 0.7 |
| 2 repeats | Poor affinity | Poor affinity |
| 1 repeat | No affinity | No affinity |

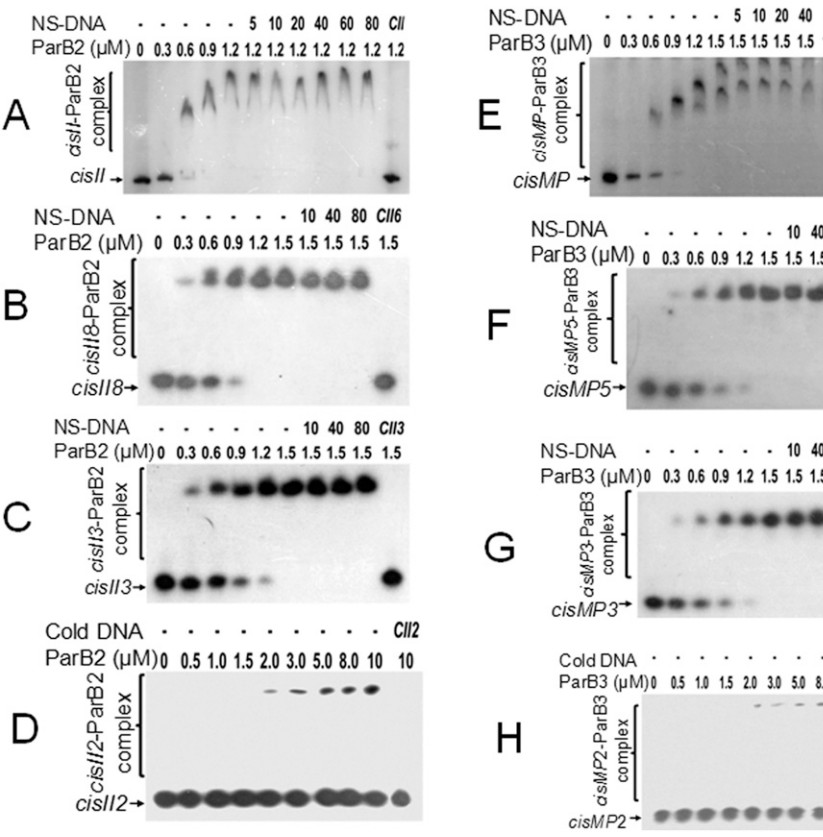

**Figure 4. ParBs interaction with *cisII* and *cisMP* and their repeats variants.**

**(A, B, C, D)** The radiolabelled full-length *cisII* containing 11 repeats (A), 8 repeats (B), 3 repeats (C), and 2 repeats (D) (Table S2) were incubated with the increasing concentration of ParB2. **(E, F, G, H)** Similarly, the radiolabelled full-length *cisMP* containing eight repeats (E), five repeats (F), three repeats (G), and two repeats (H) were incubated with the increasing concentration of ParB3. A saturating concentration of DNA–protein ratio was chased with increasing molar ratio of non-specific DNA (NS-DNA) and products were analysed on native PAGE. The fractions of DNA bound with proteins were quantified densitometrically and plotted as a function of protein concentration using GraphPad Prism 6. The Kd for the curve fitting of individual plots was determined and given in Table 2.

the copy number of chrII and MP was observed in Δ*cisII* and Δ*cisMP* cells, respectively, but the copy number of chrI did not change in any of the mutant backgrounds (Fig 5A). For instance, the average copy number of chrII reduced from 5.95 ± 0.32 to 1.55 ± 0.22 and MP was reduced from 10.85 ± 0.24 to 8.52 ± 0.81 in Δ*cisII* mutant. Similarly, the average copy number of MP was reduced from 10.85 ± 0.24 to 2.42 ± 0.35 in Δ*cisMP* mutant (Fig 5A). The deletion of *cisMP* did not affect the copy number of either chrI or chrII when compared with wild type. The reduction in ploidy of secondary genome replicons in deletion mutant of cognate *cis*-elements could be accounted for its effect on the replication of these replicons. However, a possibility of these deletions affecting DNA segregation in dividing cells cannot be ruled out. This possibility was checked by monitoring the frequency of cells conferring Kan[R] (expressed on ChrII or MP) when grown in the absence and

presence of kanamycin. In general, the growth of *cis* mutants was affected as compared with wild type and reduced further when grew in the presence of kanamycin as compared with its absence (Fig 5B). Furthermore, the Δ*cisII* and Δ*cisMP* mutants were grown for 14 h (log phase) in the presence and absence of kanamycin and total CFUs were measured in TGY agar plate in the presence and absence of antibiotic. Results showed that Kan[R] CFU was reduced nearly half (–Kan vs +Kan) in both the mutants when they were grown without selection pressure, whereas there was no change in Kan[R] CFU (–Kan vs +Kan) when these mutants were maintained with antibiotic selection (Fig 5C). Because the antibiotic is expressed on respective replicons, these results indicated that all daughter cells grown without selection pressure might have not inherited cognate replicon in *cis* mutant. Therefore, a possibility of chrI alone supporting the normal growth of this bacterium was suggested.

**Table 2. The dissociation constant (Kd) value of ParB2 and ParB3 interaction with repeats variants of *cisII* and *cisMP* elements, respectively.**

| No. of repeats | Dissociation constants (Kd) (mean ± SD, *μM*) | |
| --- | --- | --- |
| | **ParB2 with *cisII* variants** | **ParB3 with *cisMP* variants** |
| Full length | 0.40 ± 0.006 | 0.59 ± 0.06 |
| 8/5 repeats | 0.68 ± 0.06 | 0.81 ± 0.04 |
| 3 repeats | 1.06 ± 0.26 | 1.01 ± 0.15 |
| 2 repeats | Poor affinity | Poor affinity |
| 1 repeat | No affinity | Poor affinity |

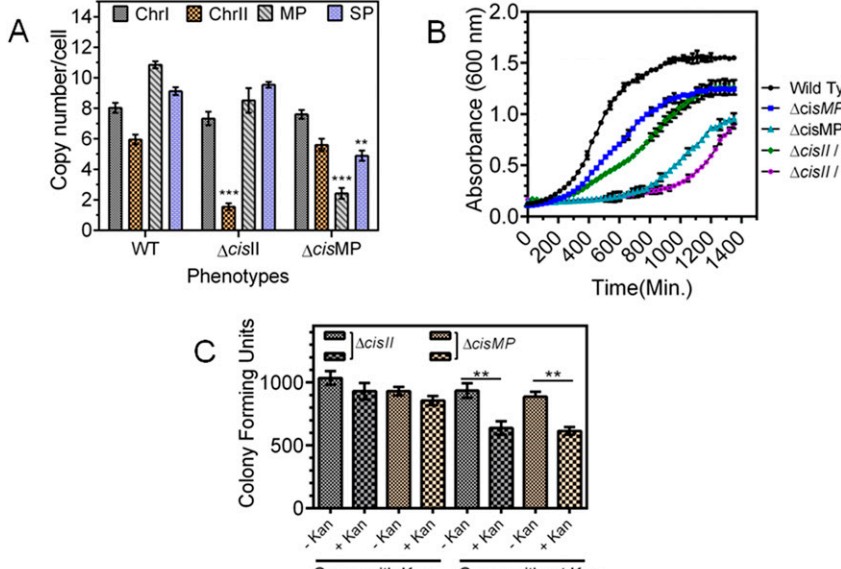

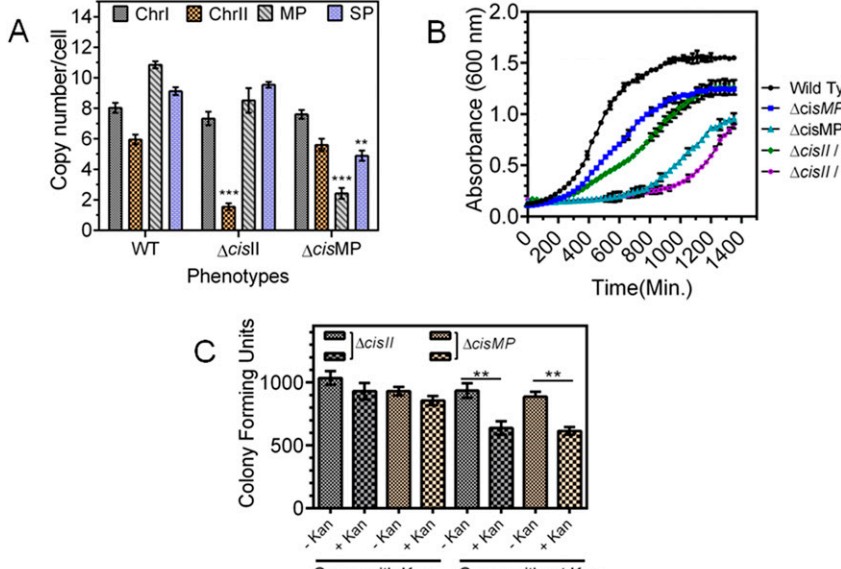

**Figure 5. Effect of *cisII* and *cisMP* deletions on genome copy number and radioresistance in *D. radiodurans*.**
**(A)** The exponentially growing *D. radiodurans* R1 (WT) and its *cisII* (Δ*cisII*) and *cisMP* (Δ*cisMP*) deletion mutants were used for the determination of copy number of each replicon (A). **(B)** Both wild type and *cis* mutants of *D. radiodurans* were checked for growth in the presence and absence of kanamycin under normal conditions (B). **(C)** These mutants were maintained in the presence (+Kan) and absence (−Kan) of antibiotics as indicated on x-axis, and the CFUs in the absence and presence of kanamycin were monitored and plotted on y-axis (C).

Similar studies have been reported in *B. subtilis* and *Mycobacterium tuberculosis* (Hassan et al, 1997; Qin et al, 1999; Richardson et al, 2016, 2019). These results together suggested that *cisII* and *cisMP* elements seem to carry functions for controlling both DNA replication and segregation in *D. radiodurans*. Furthermore, the status of chrII and MP in *cis* mutants was examined microscopically.

**cisII and cisMP deletions affected the maintenance of cognate genomic replicons**

The multipartite genome system, ploidy, and packaging of complete genetic materials into a doughnut-shaped toroidal nucleoid are considered to be the remarkable cytogenetic features in *D. radiodurans* (Harsojo & Matsuyama, 1981; Levin-Zaidman et al, 2003;

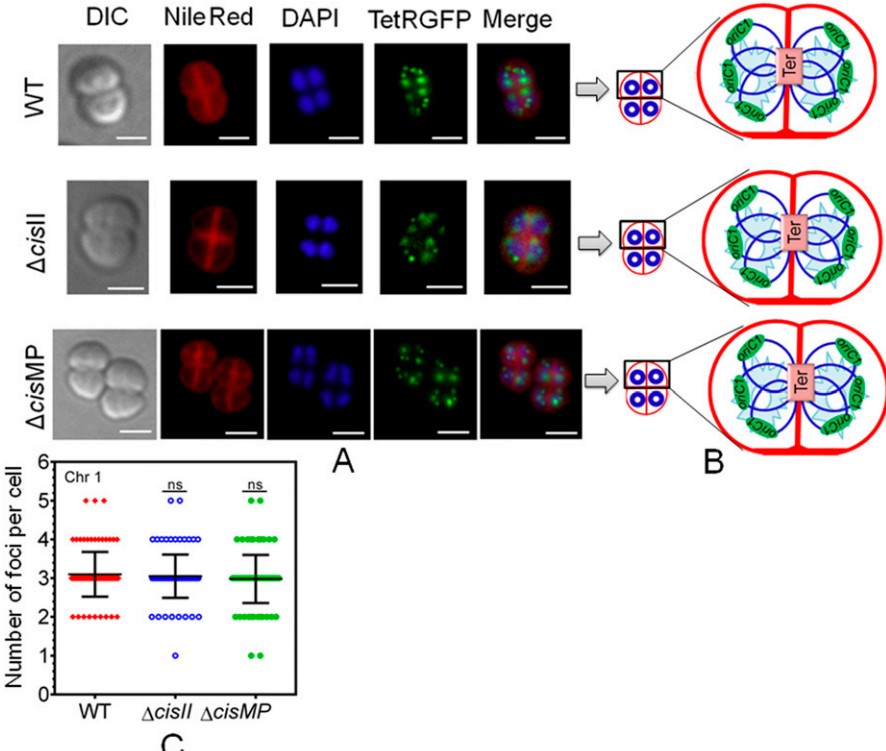

**Figure 6. The localization pattern of chromosome I in the nucleoid of wild type and secondary genome *cis* mutants of *Deinococcus radiodurans*.**
Chromosome I was tagged with *tetO*/TetR-GFP–based fluorescent reporter–operator system in both wild type (WT) and *cis* mutants (Δ*cisII* and Δ*cisMP*) as described in the Materials and Methods section and grown exponentially. **(A)** These cells were stained with Nile Red and DAPI and visualized microscopically in differential interference contrast, TRITC (Nile Red), DAPI (DAPI), and FITC (GFP) channels and images were merged (Merge) (A). The scale bar of 1 μm is used in each figure. **(B)** The schematic diagrams showing the foci position with respect to nucleoid and septum are presented for better clarity (B). **(C)** Data shown are from a single tetrad where most of these cells show a similar pattern as quantified from ~200 cells (C).

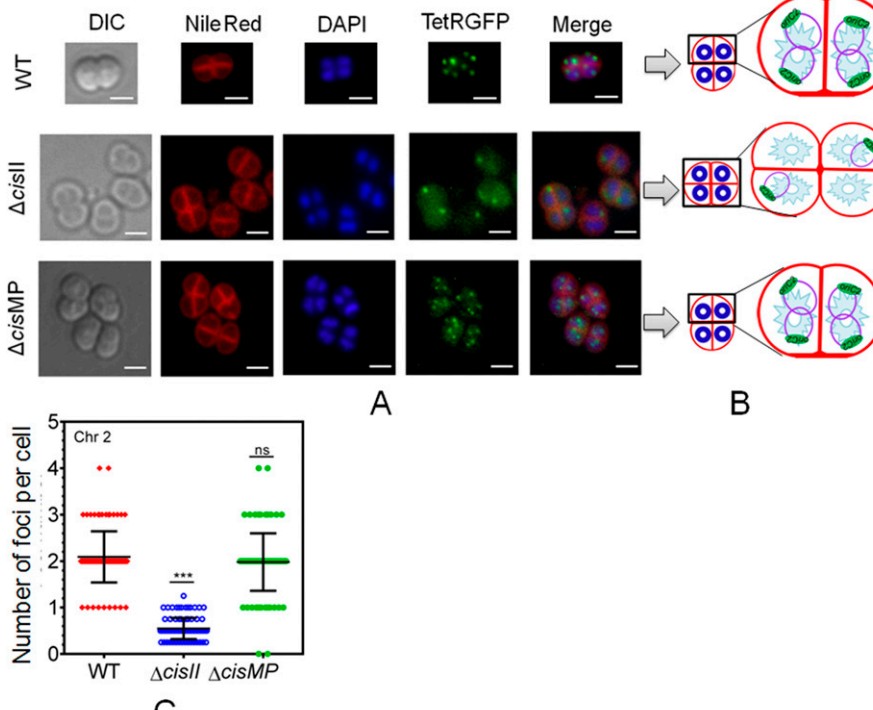

**Figure 7. The localization pattern of chromosome II in the nucleoid of wild type and secondary genome *cis* mutants of *Deinococcus radiodurans*.**
Chromosome II was tagged with a *tetO*-TetR-GFP–based fluorescent reporter–operator system in both wild type (WT) and *cis* mutants (Δ*cisII* and Δ*cisMP*) as described in the Materials and Methods section and grown exponentially. **(A)** These cells were stained with Nile Red and DAPI and visualized microscopically in differential interference contrast, TRITC (Nile Red), DAPI (DAPI), and FITC (GFP) channels and images were merged (Merge) (A). The scale bar of 1 μm is used in each figure. **(B)** The schematic diagrams showing the foci position with respect to nucleoid and septum are presented for better clarity (B). **(C)** Data shown are from a single tetrad where most of these cells show a similar pattern as quantified from ~200 cells (C).

Minsky et al, 2006). However, the organization of genomic replicons in toroidal nucleoid has not been demonstrated. In this work, we developed an FROS approach for the chrI (Fig S3A), chrII (Fig S3B), and MP (Fig S3D) in the wild-type, and Δ*cisII*- and Δ*cisMP*-mutant cells as described in the Materials and Methods section (Fig S3C and E). The exponentially growing cells were stained with Nile red and DAPI and imaged under a fluorescence microscope. Interestingly, the localization pattern of chrI in the Δ*cisII* and Δ*cisMP* mutants and wild-type cells were similar (Fig 6A). The average number of foci for chrI per cell was ~3, which were localized away from the septum and seems to be radially distributed throughout the cytoplasm in both wild type and Δ*cisII* and Δ*cisMP* cells (Fig 6B and C). A nearly similar pattern of chrI localization has been recently demonstrated in *D. radiodurans*, where the number of foci per nucleoid was shown to be in the range of 2.25–4.44 (Floc'h et al, 2019). However, when the localization pattern of chrII (Fig 7A) and MP (Fig 8A) was observed under microscope, the average number of foci for chrII (Fig 7A–C) and MP (Fig 8A–C) was found to be 2.09 and 1.1 per cell, respectively. These were too located away from the septum and spanned throughout the nucleoid in wild type. However, in the case of *cis* mutant, the localization pattern of chrII and MP was different. For instance, chrII localization in nucleoid was altered in Δ*cisII* but not in Δ*cisMP* and vice versa. Both these *cis* mutants produced a population of cells that did not contain the cognate replicon. Because the qPCR study has shown the highest number (10.85) of copies of megaplasmid per cell, the single GFP foci observed through FROS is intriguing but informative. Similar results have been reported in other limited copy plasmids such as RK2, R1, etc. in *E. coli* (Weitao et al, 2000; Pogliano et al, 2001). The decreasing number of GFP foci from 3 to 4 in ChrI to one in MP agreed with their

size difference and if that has affected resolution cannot be ruled out.

Furthermore, we studied the population dynamics of cells for the presence and absence of chrI, chrII and MP in wild type and *cis* mutants. The majority of wild type cells showed ≥3 foci corresponding to chrI, 2 foci for chrII and single focus for MP (Fig 9A). However, the number of cells with ≥3 foci corresponding to chrI in both the *cis* mutants was nearly similar to wild type. For chrII, these were nearly the same in *cisMP* mutant and wild type with two foci in each. However, in *cisII* mutant, there was hardly any cell containing two foci and all the population was divided almost equally into cells with a single focus or no focus (Fig 9B). Similarly, wild-type and *cisII* mutant showed a nearly similar number of foci corresponding to MP, which was different in *cisMP* mutant. For instance, the *cisMP* mutant showed nearly half of the cells with a single focus and the other half population did not show foci corresponding to MP (Fig 9C). These results highlighted the roles of *cisII* and *cisMP* as "*ori*" and *parS*-like functions in the secondary genome replicons of *D. radiodurans*. Furthermore, the FROS-based localization provided the first evidence that both primary and secondary genome replicons are the part of the compact nucleoid and these replicons seem to be independently maintained in this multipartite genome-harboring bacterium.

### *cisII* and *cisMP* mutants showed reduction in γ-radiation resistance

The effect of *cisII* and *cisMP* deletion on γ radiation response of *D. radiodurans* was examined. The Δ*cisII* mutant was more sensitive to γ radiation than Δ*cisMP* mutant (Fig 10). Furthermore, the mutants

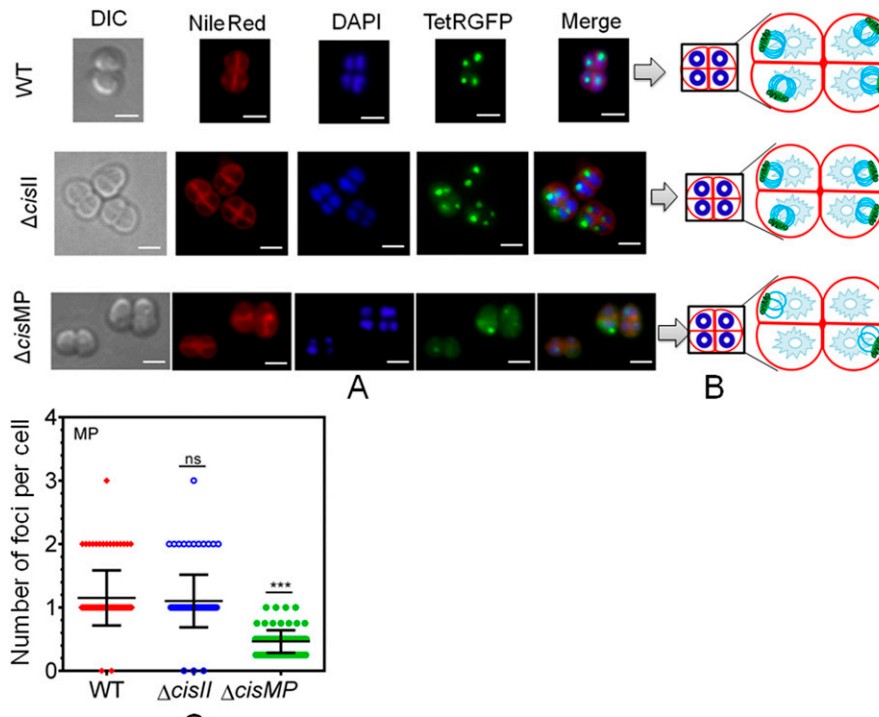

**Figure 8. The localization pattern of megaplasmid in the nucleoid of wild type and secondary genome *cis* mutants of *Deinococcus radiodurans*.**
The megaplasmid was tagged with a *tetO*-TetR-GFP–based fluorescent reporter–operator system in both wild type (WT) and *cis* mutants (Δ*cisII* and Δ*cisMP*) as described in the Materials and Methods section and grown exponentially. **(A)** These cells were stained with Nile Red and DAPI and visualized microscopically in differential interference contrast, TRITC (Nile Red), DAPI (DAPI), and FITC (GFP) channels and images were merged (Merge) (A). The scale bar of 1 μm is used in each figure. **(B)** The schematic diagrams showing the foci position with respect to nucleoid and septum are presented for better clarity (B). **(C)** Data shown are from a single tetrad where most of these cells show a similar pattern as quantified from ~200 cells (C).

that were maintained without antibiotic selection showed nearly twofold higher sensitivity to γ radiation as compared with those grown with antibiotic (Fig 10). The loss of γ radiation resistance in *cis* mutants that apparently had shown the reduction in copy number in nearly half of the population, and the other half were anucleated for cognate replicons, suggested the role of chrII and MP in radioresistance of this bacterium. Earlier, it was shown that the reduction in copy number of chrII and MP in Δ*parA2*Δ*parA3* double mutants has affected γ radiation response in *D. radiodurans* (Maurya et al, 2019a). The mechanisms responsible for the underlying loss of radioresistance in cells that have either a reduced copy number or the complete loss of chrII and MP are not understood. However, these replicons encode a large number of proteins including PprA (Narumi et al, 2004), DR_A0282 (Das & Misra, 2011), DR_A0074, and DR_A0065 (Ghosh & Grove, 2006) on chrII, and DR_B0100 (Kota et al, 2010), DR_B0090, and DR_B0091 (Desai et al, 2011), which have been characterized for their roles in DNA repair and radioresistance (Misra et al, 2013). Therefore, the reduction in the copy number or the absence of these proteins in *cis* mutants if have affected the radioresistance of *D. radiodurans* cannot be ruled out. These results suggested that chrII and MP replicons, which are perturbed in *cisII* and *cisMP* mutants, contributes to the radioresistance in this bacterium.

## Discussion

*D. radiodurans*, an extraordinary radioresistant bacterium is characterized for its peculiar cytogenetic features like the multipartite genome, ploidy, and tetrad cell morphology. Mechanisms underlying DNA replication and genome segregation have not been studied in detail in this bacterium, and very limited information is available in other MGH bacteria such as *V. cholerae*, *R. sphaeroides*, *S. meliloti*, and *D. radiodurans* (Fogel & Waldor, 2005; Charaka & Misra, 2012; Dubarry et al, 2019; Maurya et al, 2019a). Recently, the nucleoid dynamics and radial positioning of *ori* and *ter* in chrI of *D. radiodurans* have been demonstrated as a function of cell cycle (Floc'h et al, 2019). However, the organization of the secondary genome replicons in the nucleoid and their stable inheritance in dividing cells under normal and γ radiation stressed conditions had not been studied. This study has brought forth some evidences to suggest that the direct repeats present upstream to *parAB* operon in chrII (*cisII*) and MP (*cisMP*) function like "*ori*" and *parS*-like elements in *D. radiodurans*. By using FROS approach, we demonstrated the radial localization of chrI, chrII, and MP in the nucleoid and away from septum. The search of the homologues of RepC encoded on secondary genome element in *A. Tumefaciens*, *B. abortus*, *B. pseudomallei, and S. Meliloti* and RctB in *V. cholerae* did not pick up these proteins in the chrII and MP in *D. radiodurans* (data not shown). However, chrI encodes the DnaA homologue of *E. coli*. This does not rule out the possibility of some uncharacterized replication initiators for secondary genome elements in this bacterium.

We used DnaA and ParBs as probes for *ori* and *parS*-like sequences and confirmed that *cisII* and *cisMP* elements have both DnaA boxes and iterons. Our results demonstrating the (i) conversion of a non-replicative plasmid into a replicative one in the presence of these *cis*-elements, (ii) reduction in copy number of cognate replicon in absence of these elements, and (iii) an increased frequency of cells devoid of chrII and MP in respective

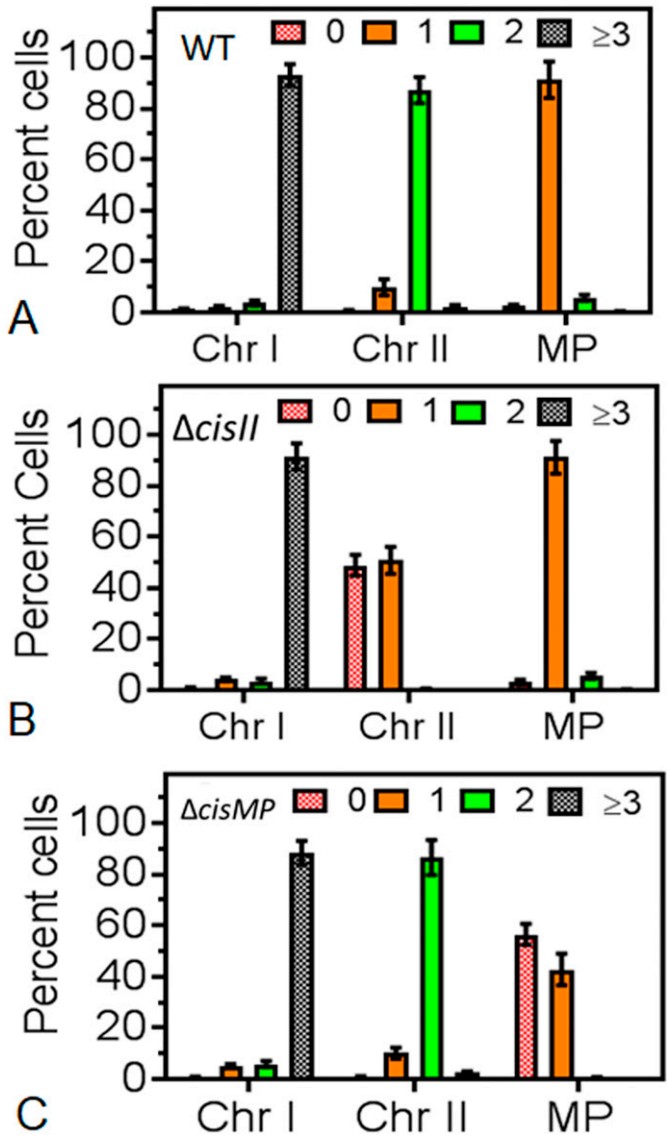

**Figure 9. Distribution of genomic replicons in wild type and *cis* mutants of *Deinococcus radiodurans* observed under fluorescence microscopy.**
**(A, B, C)** The chromosome I (ChrI), chromosome II (ChrII and megaplasmid [MP]) of wild type (A), cisII (B), and cisMP (C) mutant of *D. radiodurans* were tagged with GFP-based fluorescent reporter–operator system and cells were imaged under fluorescence microscopy as described in Figs 6–8. Nearly 200 cells of each type were analysed for the absence (0) and the presence of varying numbers (1, 2, ≥ 3) of foci corresponding to all three replicons. Data presented here are mean ± SD (n = 200).

deletion mutant, together supported the hypothesis that both *cisII* and *cisMP* elements play the important roles in the regulation of DNA replication and segregation functions in this bacterium. The presence of both "*ori*" and *parS*-like sequences in close vicinity, if helps the cells to differentially regulate the growth phase dependent interaction of replication complex with *ori* region and segregation complex with *parS*-like region cannot be ruled out. Earlier, the presence of *parS*-like sequences in proximity to "*oriC*" in their chromosomes and the molecular crosstalk between the components of the macromolecular events like replication and

segregation has also been reported in some bacteria. For instance, in *B. subtilis*, Soj (a ParA homologue) directly interacts with DnaA proteins and regulates the replication at *oriC* (Scholefield et al, 2011). In *C. crescentus*, DnaA controls chromosome segregation in ParA dependent manner as well as by binding directly to the *parS* site (Marczynski et al, 2019). Although both the chromosomes in *V. cholerae* encode separate replication and segregation components (Egan & Waldor, 2003), the interdependent regulation of these two processes has been observed. For instance, ParA1, a genome segregation protein, encoded on chrI in *V. cholerae* stimulates DNA replication through its direct interaction with DnaA while ParB2, a *parS* binding protein, encoded on chrII showed inhibition of replication initiation in chrI (Kadoya et al, 2011). On contrary, ParB2 stimulates the replication of chrII by decreasing RctB binding to inhibitory DNA sequences adjacent to the *oriII* in *V. cholerae* (Pal et al, 2005; Venkova-Canova et al, 2013). In *D. radiodurans*, the interaction of segregation proteins with replication initiation proteins (DnaA and DnaB) and an arrest of genome segregation affecting DNA replication have been demonstrated (Maurya et al, 2019b; Maurya & Misra, 2020 Preprint). Although secondary genome elements in *D. radiodurans* do not encode for any of the known DNA replication initiation proteins, and cisII and cisMP are being implied to regulate DNA replication through DnaA, the possibility of some additional uncharacterized mechanism(s) for the regulation of replication in chrII and MP cannot be ruled out.

Furthermore, we tried to distinguish the replication and segregation functions of these *cis*-elements in vivo. We could delete *cisII* and *cisMP* from cognate genome elements but not *oriCI* from chrI of this bacterium and these cells were viable under normal conditions indicating a strong possibility that chrII and MP replicons are not required for the normal growth in this bacterium. Microscopically, we observed that only nearly half of the populations of Δ*cisII* and Δ*cisMP* mutants lack cognate replicon indicating a possibility of functional redundancy of these *cis*-elements in regulation of DNA replication in chrII and MP. Because chrII was designated as chromosome because of the presence of certain essential genes as chromosomal markers, nearly no effect of *cisII* deletion on survival of *D. radiodurans* is intriguing. However, the growth rates of these mutants under normal conditions and in the absence of antibiotic selection were different from the wild type. Results supported the hypothesis where we observed (i) the reduction in copy number of cognate replicons in mutant grown under antibiotic selection and (ii) the sizable cell populations of mutants did not have cognate replicons as monitored by the absence of antibiotic resistance encoded on them as well as microscopically. We argued that *cis* mutants, which were maintained without antibiotic selection may have grown as the heterogeneous population having cells with and without cognate replicons. When these are selected for Kan[R], nearly half of the population was found to be Kan[S] (Fig 5B and C). Presumably, these were the ones that have survived with the help of chromosome I and were devoid of secondary replicons because of segregation defects. Although mutational analysis of these repeats and their interaction with cognate ParBs has provided a strong evidence on the role of these elements in genome segregation, loss of chrII and MP in nearly 50% population of *cisII* and *cisMP* mutants, respectively, also supports the role of these *cis*-elements in genome segregation.

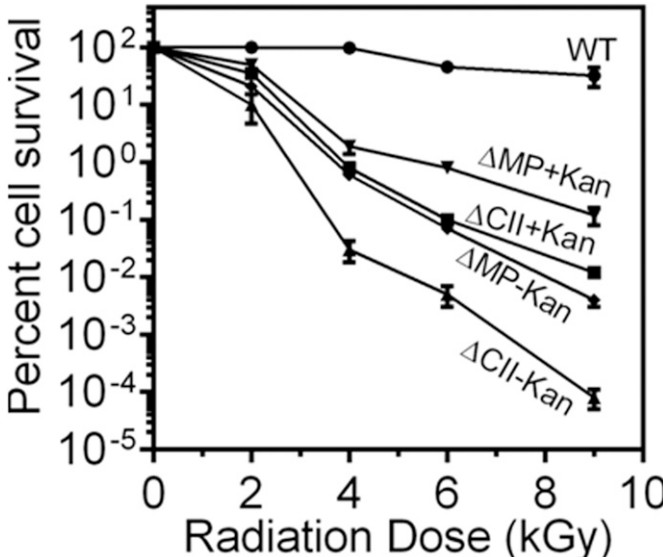

**Figure 10. Gamma radiation response of *cisII* and *cisMP* mutants of *Deinococcus radiodurans*.**
The γ radiation response of wild type (WT) and its *cisII* (ΔCII) and *cisMP* (ΔMP) mutants grown in the presence (+Kan) and absence (–Kan) of kanamycin was monitored. Data presented here are mean ± SD (n = 6) from the reproducible experiments.

Furthermore, the FROS results where 2–3 cells in tetrad were devoid of foci in respective replicons supported the argument of segregation defect in cells lacking *cis*-elements. Survival of other three cells in tetrad without replicons conferring antibiotic resistance under selection pressure is intriguing and a strong possibility of these supported by the neighbouring antibiotic-resistant cells cannot be ruled out. The careful monitoring of the localization pattern of chrI, chrII, and MP in nucleoid of wild type and *cis* mutants yielded very interesting results. The localization pattern of chrI was the same in both wild type and *cisII* and *cisMP* mutants, and agrees with the recent findings reported independently (Floc'h et al., 2019). Also, there was no change in the ploidy status of chrI in mutant cells. On the other hand, the localization pattern of chrII and MP in the nucleoid as well as the ploidy status of these replicons in cells deleted for *cisII* and *cisMP* elements, respectively, were grossly altered. Thus, we provided evidence that all three replicons are packaged in a single nucleoid and the "*ori*" region of these replicons is excluded from the membrane-proximal region of the cell. The anchoring of the chromosome to membrane-proximal region through SMC proteins has been reported in other bacteria (Ben-Yehuda et al, 2005; Thanbichler & Shapiro, 2006; Yamaichi et al, 2012). It would be worth mentioning that ParAs encoded on ChrII and MP are found to be functionally redundant (Maurya et al, 2019a), while ParB2 and ParB3 are not (Maurya et al, 2019b), and now *cis*-elements are differentially regulating both replication and segregation of cognate replicons in this bacterium.

In conclusion, for the first time, we have characterized the *ori* and *parS*-like regions in *cisII* and *cisMP* elements in chrII and MP and showed that these *cis*-elements regulate replication and segregation functions in an MGH bacterium *D. radiodurans*. Furthermore,

we provided evidence that the maintenance of these secondary replicons is independent of the primary chromosome, yet the chrI, chrII, and MP are packaged in form of a compact nucleoid. Last, this study also provides evidence that the secondary genome replicons have roles in the extraordinary radioresistance of this bacterium. Recently, many bacteria have been discovered with a multipartite genome system, and surprisingly the most of these are pathogenic/inhabitant to some forms of life. Regulation of origin of replication, incompatibility and copy number as understood by taking *E. coli* as model system, would be worth revisiting in the context of ploidy and multipartite genome system as known in many MGH bacteria. This study may serve as an example and provide starting materials for answering many of these questions under newly born multipartite genome biology in bacteria. Apart from this, the independent studies would be undertaken to understand (i) the dynamics of the multipartite genome during γ radiation recovery and normal cell cycle regulation in this bacterium, and (ii) DNA damage responsive territorial change of genomic replicons in cells recovering from γ radiation exposure and its impact on genome functions in *D. radiodurans*. The presence of chrII and MP in certain population of *cisII* and *cisMP* mutants, respectively, does not rule out a strong possibility of DnaA and *cisII* or cisMP independent regulation of replication in these replicons and would be worth investigating independently.

## Materials and Methods

### Bacterial strains, plasmids, and materials

*D. radiodurans* R1 (ATCC13939) was grown in TGY (1% tryptone, 0.1% glucose, and 0.5% yeast extract) medium at 32°C. *E. coli* strain NovaBlue was used for cloning and maintenance of all the plasmids, whereas *E. coli* strain BL21 (DE3) pLysS was used for the expression of recombinant proteins. The pNOKOUT plasmid (a non-replicative plasmid in *D. radiodurans*) was used for generating gene knockouts and bioassay of *cis*-elements for *origin of replication* function in *D. radiodurans*. Standard protocols for all the recombinant techniques were used as described earlier (Green & Sambrook, 2012). Molecular biology grade chemicals and enzymes were procured from Merck Inc. and New England Biolabs. Radiolabeled nucleotides were purchased from the Board of Radiation and Isotope Technology, Department of Atomic Energy, India (BRIT). All the oligonucleotides used in this study are listed in Table S1 and all bacterial strains and plasmids are listed in Table S2.

### Bioinformatic analysis

The nucleotide sequences close to the origin region and upstream to *parAB* operons in the circular map of ChrII (99–544 bp region) and MP (177,446–417 bp region) were analysed for potential repeats and consensus sequence using Mellina II web tool (Okumura et al, 2007) and WebLogo online tool (Crooks et al, 2004) and tentatively named as *cisII and cisMP*, respectively. The putative DnaA boxes in *cisII* were predicted using DOriC database (DoriC accession number– ORI10030006) (Luo & Gao, 2019). The DnaA boxes in *cisMP* were

marked based on sequence homology with known DnaA boxes from other bacteria.

## Cloning of *cis*-elements and their maintenance

The *cisII* and *cisMP* were PCR-amplified from genomic DNA of *D. radiodurans* using cisIIFw/cisIIRw and cisMPFw/cisMPRw primers, respectively (Table S1). The *cisII* was cloned at *Xba*I site, whereas *cisMP* at *Apa*I and *Eco*RI sites in pNOKOUT (Khairnar et al, 2008) to yield pNOKcisII and pNOKcisMP plasmids, respectively (Table S2). These plasmids were transformed in both wild type and *recA* mutant of *D. radiodurans* and transformants were scored at kanamycin (6 μg/ml) containing TGY agar plates. For plasmid maintenance, the transformants were streaked on kanamycin containing TGY agar plates. The bacterial growth was also monitored in TGY liquid medium in the presence of antibiotic as required, at 32 °C and OD 600 nm was monitored using Synergy H1 Hybrid Multi-mode microplate reader, Bio-Tek. The maintenance of pNOKcisII and pNOKcisMP plasmids in *D. radiodurans* was confirmed with antibiotic resistance as well as by isolating these plasmids from transformant as described earlier (Meima & Lidstrom, 2000). The isolated plasmid DNA was digested with *Xba*I in the case of pNOKcisII and *Apa*I and *Eco*RI in the case of pNOKcisMP, and the restriction digestion pattern was compared with the original plasmid on the agarose gel.

## Protein purification

The recombinant DnaA of *D. radiodurans* expressing on pETDnaA in *E. coli* BL21 (DE3) pLysS was purified by Ni-NTA affinity chromatography as described earlier (Maurya & Misra, 2020 Preprint). In brief, *E. coli* cells carrying pETDnaA were induced with 0.5 mM isopropyl-β-D-thio-galacto-pyranoside (IPTG) and cell pellet was suspended in buffer A (20 mM Tris–HCl, pH 7.6, 300 mM NaCl and 10% glycerol), 0.5 mg/ml lysozyme, 1 mM PMSF, 1 mM MgCl₂, 0.05% NP-40, 0.05% Triton X-100, and protease inhibitor cocktail. The resultant mixture was sonicated on ice and cell lysate was centrifuged at 13,500*g* for 30 min at 4°C and the clear supernatant was purified through Ni-NTA column chromatography (GE Healthcare). The recombinant ParBs encoded on chromosome II (ParB2) and megaplasmid (ParB3) in *D. radiodurans* were purified from recombinant *E. coli* using Ni-NTA affinity chromatography (Maurya et al, 2019b). Fractions with >95% purity were pooled and purified from HiTrap Heparin HP affinity columns (GE Healthcare Life Sciences) using a linear gradient of 20–300 mM NaCl, followed by precipitation with 30% wt/vol ammonium sulphate at 8°C, and molecular exclusion chromatography using the standard protocol. Histidine-tag was removed from the recombinant proteins by digesting with Factor Xa protease followed by passing through Ni affinity column. The absence of his-tag in the protein eluted in flow-through fraction was confirmed using poly histidine antibodies and the concentration was determined by taking OD at 280 nm in NanoDrop (Synergy H1 Hybrid Multi-Mode Reader Biotek).

## DNA–protein interaction studies

DNA binding activity of DnaA, ParB2, and ParB3 was monitored using electrophoretic mobility shift assay as described earlier (Charaka & Misra, 2012; Maurya et al, 2019a) with different versions of *cis*-element as DNA substrates. In brief, the full-length *cisII* and *cisMP* and their repeat derivatives (Fig S4A and B) were PCR-amplified using sequence-specific primers (Table S1) except the two repeats and one repeat that were chemically synthesized and annealed with the complementary strand. The [³²P] labeled DNA probes were used for DNA binding assay. For the competition assay, saturating concentrations of DnaA or ParBs with radiolabelled DNA substrates were chased with different concentrations of unlabeled non-specific DNA (NS-DNA) of similar length (Table S1) and a 10-fold higher concentration of same unlabeled *cis* sequence. The products were resolved on 6–10% native PAGE gel as required. The DNA band intensity was determined densitometrically using Image J software, and the fractions of bound DNA were plotted against different protein concentrations using GraphPad Prism 5. The Kd value for the curve fitting of individual plots was determined.

## Construction of deletion mutants and cell survival studies

The constructs for generating knockout of both *cis*-elements were made in pNOKOUT plasmid as described earlier (Khairnar et al, 2008). In brief, ~500 bps upstream and ~500 bps downstream fragments to the *cisII* and *cisMP* were PCR amplified with the primers (Table S1) and cloned at *Kpn*I/*Eco*RI and *Bam*HI/*Xba*I sites in pNOKOUT plasmid to yield pNOKCII and pNOKCM plasmids, respectively (Table S2). These plasmids were linearized with *Xmn*I and transformed into *D. radiodurans* cells. The homozygous replacement of *cisII* and *cisMP* by *nptII* was achieved through several rounds of sub-culturing and then ascertained by diagnostic PCR using flanking primers (Table S1 and Fig S2A–C). The confirmed mutant cells were maintained in the absence of selection pressure but ensured for antibiotic resistance before the experiments.

The survival of Δ*cisII* and Δ*cisMP* deletion mutants was monitored under normal conditions and in response to different doses of γ radiation as described earlier (Misra et al, 2006). In brief, the cells were grown in the absence and presence of antibiotics. These cells were plated on TGY agar plate with and without antibiotics under normal conditions. These cells were independently exposed to different doses of γ-radiation (2, 4, 6, and 9 kGy) at the dose rate of 1.81 kGy/h using Gamma Cell 5000 irradiator (⁶⁰Co; Board of Radiation and Isotopes Technology, DAE). The equal number of irradiated cells and their SHAM controls were washed in PBS and the different dilutions were plated in replicates on TGY agar plates with antibiotics, as required. The plates were incubated at 32°C for 45–48 h and CFUs were counted.

## Estimation of genome copy number

The copy number of genomic replicon and mini replicons like pNOKcisII and pNOKcisMP was determined in the exponentially growing *D. radiodurans* cells using modified protocols of Breuert et al (2006) as described earlier in Maurya et al (2019a). In brief, the density and cell numbers of the independent sample were

normalized for a fixed OD 600 nm and by CFUs. These cells lysed in buffer containing 10 mM Tris, pH 7.6, 1 mM EDTA, and 4 mg/ml lysozyme followed by repeated cycles of freezing and thawing. The clear cell-free extract was used for quantitative Real-Time PCR. The qPCR analysis involves three different genes of similar PCR efficiency from each genomic replicon of *D. radiodurans*. For instance, *ftsE*, *ftsK*, and *ftsZ* for chromosome I; DR_A0155, DR_A0182, and *pprA* for chromosome II; DR_B0003, DR_B0030, and DR_B0104 for megaplasmid and DR_C0001, DR_C0018, and DR_C0034 for small plasmid (Table S1). For mini replicons, the *nptII* (Kan$^R$) and *bla* (Amp$^R$) genes of plasmid backbones (Table S1) were used for copy number determination. The 300 bp PCR amplified DNA fragments were quantified by nanodrop and DNA concentrations were calculated using the molecular mass computed with "oligo calc" (www.basic.northwestern.edu/biotools). A serial dilution was made for each standard fragment and qPCR was carried out by following the MIQE guidelines using Roche Light cycler (Bustin et al, 2009) and the cycle threshold (Ct) values were determined. The copy number of each replicon is quantified by comparing the results with a standard plot. The average copy number reflected from all three genes per replicon was analysed using statistical analysis of the data obtained from the three independent biologic replicates for each sample.

## Construction of *tetO* insertion derivatives of the genomic replicons in *D. radiodurans*

The 240 repeats of *tetO* operator cassette from pLAU44 plasmid (Lau et al, 2003) were integrated near the putative "*origin of replication*" at 1.5° in ChrI, 4° in ChrII, and 4.5° in MP of the circular map of these replicons in *D. radiodurans*. For that, the PCR amplifications of 10,713–11,715 bp region corresponding to 1.5° of ChrI using ChrI(1.5°)Fw and ChrI(1.5°)Rw primers, 4659–c5691 bp region corresponding to 4° of ChrII using ChrII(4°)Fw and ChrII(4°)Rw primers, and 2,203–3,000 bp region corresponding to 4.5° of MP using Mp (4.5°)Fw and Mp(4.5°)Rw were carried out. These PCR fragments were cloned in pLAU44 at *Xba*I-*Sca*I sites to yield p44Ch1, p44Ch2 and p44MP plasmids having homologous sequences of ChrI, ChrII, and MP, respectively. Furthermore, an expressing cassette of spectinomycin resistance gene was amplified from pVHS559 (Charaka & Misra, 2012) using SpecFw and SpecRw primers and cloned at *Nhe*I-*Xho*I sites in p44Ch1, p44Ch2, and p44MP to give p44SCh1, p44SCh2, and p44SMP, respectively. These constructs were confirmed by restriction digestion, PCR and DNA sequencing. The p44SCh1, p44SCh2, and p44SMP plasmids were separately transformed into wild type, and Δ*cisII* and Δ*cisMP* mutants of *D. radiodurans*. The transformants were scored under antibiotics selection and the homogenous insertion of *tetO* repeats in the genome was achieved through several rounds of subculturing. This was ascertained by diagnostic PCR using different sets of primers (Table S1). The resultant strains of *D. radiodurans* were named with suffix as R1::ChrI-*tetO*, R1::ChrII-*tetO*, R1::MP-*tetO*, Δ*cisII*::ChrI-*tetO*, Δ*cisII*::ChrII-*tetO*, Δ*cisII*::MP-*tetO*, Δ*cisMP*::ChrI-*tetO*, Δ*cisMP*::ChrII-*tetO*, and Δ*cisMP*::MP-*tetO* as listed in Table S2.

## Construction of TetR-GFP expression plasmid in *D. radiodurans*

The coding sequence of *tetR* was PCR amplified from pLAU53 (Lau et al, 2003) using gene-specific primers (Table S1) and cloned at *Sac*I and *Sal*I sites in pDSW208 (Weiss et al, 1999) to give pDTRGFP plasmid. Furthermore, *tetR-gfp* was PCR amplified from pDTRGFP plasmid using TetRsclApIFw and GFPXbaIRw primers and cloned in pRADgro (Khairnar et al, 2008) at *Apa*I-*Xba*I sites to produce pRADTRGFP (Table S2). The constitutive expression of TetR-GFP from pRADTRGFP plasmid in *D. radiodurans* was confirmed by fluorescence microscopy and the Western blotting using anti-GFP antibodies (Fig S5A and B). The *tetO* operator containing cells of *D. radiodurans* R1, Δ*cisII* and Δ*cisMP* mutants (Table S2) expressing TetR-GFP on pRADTRGFP plasmid were observed under a fluorescence microscope as described earlier (Maurya et al, 2019a). In brief, overnight (14 h) grown cultures were harvested, washed with PBS and stained with DAPI (4′,6-diamidino-2-phenylindole, dihydrochloride; 0.2 mg/ml) for nucleoid and Nile red (1 mg/ml) for membrane. These samples were imaged through differential interference contrast, DAPI, FITC, and TRITC channels under an inverted fluorescence microscope (Olympus IX83) fitted with an Olympus DP80 CCD monochrome camera. The images were aligned and processed using in-built software "CellSens." A population of ~200 cells was analysed for results and data were presented as a scattered plot. One representative image belonging to more than 90% population is presented separately as differential interference contrast, DAPI, Nile Red, and GFP and merged of all these. Experiments were repeated independently to ensure the reproducibility and significance of these data.

# Data Availability

This article contains data that have been recorded using low-throughput technologies and are available anytime for cross references.

# Supplementary Information

# Acknowledgements

The authors are grateful to Dr SK Apte for his comment on the work and results, and Dr Swathi Kota, Dr YS Rajpurohit, and Dr Sheetal Uppal for their technical inputs. The authors also thank Ms Shruti Mishra and Ms Reema Chaudhary for reading through the manuscript, their comments, and help in fluorescence imaging. GK Maurya is grateful to the Department of Atomic Energy, India, for the research fellowship.

## Author Contributions

GK Maurya: conceptualization, resources, data curation, formal analysis, investigation, methodology, and writing—original draft, review, and editing.

HS Misra: conceptualization, resources, data curation, formal analysis, supervision, funding acquisition, validation, investigation, visualization, methodology, project administration, and writing—original draft, review, and editing.

## Conflict of Interest Statement

The authors declare that they have no conflict of interest.

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
