## [Reviewer comments · Life Science Alliance]

Life Science Alliance

Characterization of ori and parS-like functions in secondary genome replicons in *Deinococcus radiodurans*

Ganesh Maurya & Hari Misra

DOI: <https://doi.org/10.26508/lsa.202000856>

Corresponding author(s): Dr. Hari Sharan Misra, Bhabha Atomic Research Centre

Review Timeline:

Submission Date:	2020-07-22
Editorial Decision:	2020-08-20
Revision Received:	2020-10-26
Editorial Decision:	2020-11-02
Revision Received:	2020-11-04
Accepted:	2020-11-04

Scientific Editor: Shachi Bhatt

Transaction Report:

August 20, 2020

Re: Life Science Alliance manuscript #LSA-2020-00856-T

Dr. Hari S Misra
Bhabha Atomic Research Centre
Molecular Biology Division
Trombay
Mumbai, Maharashtra 400085
India

Dear Dr. Misra,

Thank you for submitting your manuscript entitled "Characterization of ori and parS-like functions in secondary genome replicons and its maintenance independent of chromosome I in *Deinococcus radiodurans*" to Life Science Alliance (LSA). The manuscript has been reviewed by the editors and outside referees (reviewer comments below). As you will see, the reviewers were very enthusiastic about the study and its potential impact, and have raised only minor concerns that should be addressed prior to further consideration of the manuscript at LSA. Therefore, although we are unable to publish the current version of the manuscript, we would kindly encourage you to submit a revised version that addresses the referees' concerns.

We would be happy to discuss the individual revision points further with you should this be helpful. The typical timeframe for revisions is three months. Please note that papers are generally considered through only one revision cycle, so strong support from the referees on the revised version is needed for acceptance. When submitting the revision, please include a letter addressing the reviewers' comments point by point.

Thank you for this interesting contribution to Life Science Alliance. We are looking forward to receiving your revised manuscript.

Sincerely,

Shachi Bhatt
Executive Editor
Life Science Alliance

B. MANUSCRIPT ORGANIZATION AND FORMATTING:

Reviewer #1 (Comments to the Authors (Required)):

This study examines chromosome maintenance in *Deinococcus radiodurans*, a bacterial species with multiple chromosomes/plasmids. Our knowledge of chromosome maintenance is largely derived from species with single chromosomes, and there is significant recent interest in how this process is regulated in organisms with multiple chromosomes. *D. radiodurans* has two large chromosomes, Chr1 and Chr2, a "megaplasmid" (MP) and a smaller plasmid. Specifically, this study addresses the role of arrays of repeat sequences on Chr2 and the megaplasmid (MP), which the authors call *cisII* and *cisMP*. They show that both DnaA, the replication initiator (of Chr1), and the ParBs (ParB-II and ParB-MP), bind specifically to these repeats *in vitro*. When the arrays are deleted, the respective chromosome is unstable, measured via copy number and fluorescence tagging approaches. Chromosome loss damages the species resistance to radiation, a property of this species that has made it of high general interest. The arrays can confer the ability to replicate to a plasmid without a *D. radiodurans* origin of replication. These data suggest that the arrays contain the functions of the origin of replication and partition site, *parS*. The experiments are well-designed and controlled. I do

have some concerns about the interpretation of origin function and the conclusion about segregation, but I agree that the in vitro and in vivo data that DnaA and ParBs participate via binding these repeats are convincing.

1. The authors avoid any discussion of replication initiators of these *D. radiodurans* secondary chromosomes, and focus only on DnaA. In other multi-chromosomal species such as *V. cholera*, the origin of replication of Chr2 uses a separate initiator (as the authors mention) although they also often also employ DnaA.

a) The ability to delete the arrays is problematic if they represent the origin of replication, which would be essential for DNA replication of the affected chromosome, and so every and all daughter cells would lack this chromosome. Since this is not the case, the data indicate that these are accessory sequences rather than core sequences, and the latter are acted upon by another protein. What are the putative initiators for Chr2 and MP? Are there homologues of other plasmid-like initiators, for example? The authors should discuss why it might be possible to delete these arrays in the first place.

b) Are the entire arrays (all 11 copies) deleted in the chromosomal deletions?

c) Figs 7 and 8: how many cells have no copies of the affected chromosome? It is not possible to determine from the graphs (as drawn, it looks like no cells with 0 chromosomes). This number should be reported and is key to this discussion. Because the wild-type copy numbers of these chromosomes are 6-10 (from Fig 5A), it is possible that the chromosomes are essential as long as their copy number is 1 or greater. In other words, damaging replication or partition still allows cell growth as long as the chromosome is present, although lower copy number decreases gene dosage of important genes and affects growth rate/radiation resistance.

d) Fig 5B: How many copies of the chromosome are necessary to confer resistance to kanamycin? It is formally possible that a copy number of one would give Kan-sensitivity at the concentration of kanamycin used. In this case, the differences +/- kan could be explained as growth of cells with fewer copies rather than with no copies.

2. pg 17: The data do not allow the authors to conclude that there is a segregation role for the repeats. Either replication or segregation defects due to the deletions could account for chromosome and copy number loss, but I agree that the plasmid experiments support a replication role for the repeats. Mutations that damage ParB but not DnaA binding, or mutation of parB, for example, would be necessary to make specific conclusions about segregation. The statement that "the direct repeats function like ori and parS-like elements" should be clarified.

3. pg 1: The TGS (tripartite genome segregation), or partition, system information is mis-stated and/or out of date. They are not necessarily "mostly" used, but they do contribute, typically in conjunction with SMC-like proteins and other systems. In addition, the push/pull models based on polymerization and depolymerization are not accepted for the ParABS chromosomal and plasmid systems; rather they refer to plasmid actin and tubulin-like partition systems. Although there is still debate as to exact details, the ParABS systems work differently.

4. Figs 3 & 4: What is the "nsDNA" used in these experiments, and how is "molar" ratio determined?

Reviewer #2 (Comments to the Authors (Required)):

Maurya and Misra extend this laboratory's previous characterization of genome replication in *Deinococcus radiodurans*. In this manuscript, they define trans and cis functions needed for the coordinate replication and segregation of Chromosome II and the mega-plasmid. They convincingly demonstrate that sequences upstream of the parAB loci on each genetic element (designated cisII and cisMP) are required bind to DnaA and their cognate ParB proteins and they provided inferential evidence that these elements are necessary for appropriate replication and segregation of the elements during growth. In general, the authors' conclusions are supported by the data provided. However, I have reservations concerning some of the authors' interpretations as indicated below.

Comments

1. I am concerned by the following statement: "A majority of Δ cisII and Δ cisMP cells showed the loss of chrII and MP, respectively, while chr I localization pattern in these cells remained unperturbed when compared with wild type" and others like it in the manuscript. It seems that the authors are suggesting that chrII and the MP are dispensable and that the species does not require any of the genes for survival. Is this what is being implied? The authors thoughts on this are not clear. Also, it is not obvious why the authors believe a majority (and not all) Δ cisII and Δ cisMP cells show loss of chrII and MP. Given the role and importance they ascribe to these sequences, it would seem all cells would be affected.

2. I am unsure about the authors description and explanation for the ionizing radiation sensitivity of the Δ cisII and Δ cisMP cells. The authors state, "Since, the deletion of these cis elements had caused reduction in the copy number of cognate replicon and affected the distribution of the secondary replicons in daughter cells, the loss of gamma radiation resistance could be implicated to either reduction in copy number or complete loss of secondary genome replicons in these mutants." This statement indicates, rightly so, that loss of these replicons results in sensitivity. It seems this would be the consequence of the loss of know gene functions associated with chrII and MP. This possibility is not discussed extensively, leading me to wonder if the authors are implying another more obtuse mechanism responsible of loss of radioresistance.

3. Finally, I am struck by the following statement. "Survival of other three cells in tetrad without replicons conferring antibiotic resistance under selection pressure is intriguing and a strong possibility of cross protection from the cell conferring antibiotic resistance cannot be ruled out." This suggest a fascinating possibility - one not mentioned by the authors - that there is an intercommunication between cells carrying the missing replicons and those that are not. I should like the authors to consider the fact that there is a period in which segregating cells are in communication. Long-lived proteins made in a cell with the replicons could act in a cell without the replicons, passing through the septal annulus as the cells divide.

Dear Prof. Bhatt,

Thank you so much for the review of this paper by the experts in the field and more so that they have also been curious to know more and more about the regulation of genome maintenance in this multipartite genome harboring and radiation resistant bacterium *Deinococcus radiodurans*. I personally have enjoyed reviewers' comments and suggestions that have helped us in further improvement of the presentation of this manuscript. We have answered nearly all the concerns of both the reviewers by working around their comments both experimentally and editorial rebuttals. Hope you all find revision suitable for publication of this manuscript in LSA.

(H S Misra)

On behalf of co-authors)

Reviewer #1 (Comments to the Authors (Required)):

This study examines chromosome maintenance in *Deinococcus radiodurans*, a bacterial species with multiple chromosomes / plasmids. Our knowledge of chromosome maintenance is largely derived from species with single chromosomes, and there is significant recent interest in how this process is regulated in organisms with multiple chromosomes. *D. radiodurans* has two large chromosomes, Chr1 and Chr2, a "megaplasmid" (MP) and a smaller plasmid. Specifically, this study addresses the role of arrays of repeat sequences on Chr2 and the megaplasmid (MP), which the authors call cisII and cisMP. They show that both DnaA, the replication initiator (of Chr1), and the ParBs (ParB-II and ParB-MP), bind specifically to these repeats in vitro. When the arrays are deleted, the respective chromosome is unstable, measured via copy number and fluorescence tagging approaches. Chromosome loss damages the species resistance to radiation, a property of this species that has made it of high general interest. The arrays can confer the ability to replicate to a plasmid without a *D. radiodurans* origin of replication. These data suggest that the arrays contain the functions of the origin of replication and partition site, parS. The experiments are well-designed and controlled. I do have some concerns about the interpretation of origin function and the conclusion about segregation, but I agree that the in vitro and in vivo data that DnaA and ParBs participate via binding these repeats are convincing.

Authors response: Thank you so much for recognizing the importance of this study and your encouraging words. We are grateful to you for comments asking for finer details of our findings and kind helps for moderating conclusions. These all have helped in further improvement of this

manuscript. We are happy to provide the pointwise details to your queries as below. Hope you agree.

1. The authors avoid any discussion of replication initiators of these *D. radiodurans* secondary chromosomes, and focus only on DnaA. In other multi-chromosomal species such as *V. cholera*, the origin of replication of Chr2 uses a separate initiator (as the authors mention) although they also often also employ DnaA.

Author's response: Thank you so much. This has been most obvious question for us too, and we did analysis to find out some information on this aspect. We searched the *D. radiodurans* genome for replication initiators like RepC, RctB like proteins and RepN using BLASTP. We used the RepC protein from secondary genome of *Brucella abortus*, *Burkholderia pseudomallei*, *Shinorhizobium meliloti* and *Agrobacterium tumifeciens*, RctB from Chromosome 2 of *Vibrio cholera* and RepN from *Bacillus* and *Staphylococcus aureus* for BLASTP analysis in *D. radiodurans*. The analysis suggests that *D. radiodurans* does not encode any of these known replication initiators other than DnaA (belonging to Chromosome I). Therefore, we did all our studies for identification of cis elements using DnaA as probe. In response to your comments, these points have now been discussed in revised manuscript. The snapshots of the analyses are shown below for review purpose only.

1. Compared with RepC of *Burkholderia pseudomallei*

BLAST » blastp suite » results for RID-NPUP4RWD014 Home Recent Results Saved Strategies Help

← Edit Search Save Search Search Summary How to read this report? BLAST Help Videos Back to Traditional Results Page

! Your search is limited to records that include: *Deinococcus radiodurans* R1 (taxid:243230)

Job Title	RepC (Burkholderia pseudomallei)
RID	NPUP4RWD014 <small>Search expires on 09-12 20:17 pm</small> Download All
Program	Citation
Database	nr See details
Query ID	lc Query_78911
Description	RepC (Burkholderia pseudomallei)
Molecule type	amino acid
Query Length	287
Other reports	?

Filter Results

Percent Identity	E value	Query Coverage
<input type="text"/> to <input type="text"/>	<input type="text"/> to <input type="text"/>	<input type="text"/> to <input type="text"/>
		Filter Reset

! No significant similarity found. For reasons why, click here

2. Compared with RepC of *Brucella abortus*

< Edit Search Save Search Search Summary ▾

? How to read this report? ▶ BLAST Help Videos ↶ Back to Traditional Results Page

i Your search is limited to records that include: *Deinococcus radiodurans* R1 (taxid:243230)

Job Title	RepC (Chr 2; Brucella abortus)
RID	NPUV9BDH016 <small>Search expires on 09-12 20:19 pm</small> Download All ▾
Program	? Citation ▾
Database	nr See details ▾
Query ID	lcl Query_78926
Description	RepC (Chr 2; Brucella abortus)
Molecule type	amino acid
Query Length	397
Other reports	?

Filter Results

Percent Identity	E value	Query Coverage
<input type="text"/> to <input type="text"/>	<input type="text"/> to <input type="text"/>	<input type="text"/> to <input type="text"/>
		Filter Reset

A No significant similarity found. For reasons why, click here

3. Compared with RctB of *Vibrio cholerae*
< Edit Search Save Search Search Summary ▾

? How to read this report? ▶ BLAST Help Videos ↶ Back to Traditional Results Page

i Your search is limited to records that include: *Deinococcus radiodurans* R1 (taxid:243230)

Job Title	RctB (Chr 2; Vibrio cholerae)
RID	NPUHM3ZB014 <small>Search expires on 09-12 20:14 pm</small> Download All ▾
Program	? Citation ▾
Database	nr See details ▾
Query ID	lcl Query_3101
Description	RctB (Chr 2; Vibrio cholerae)
Molecule type	amino acid
Query Length	216
Other reports	?

Filter Results

Percent Identity	E value	Query Coverage
<input type="text"/> to <input type="text"/>	<input type="text"/> to <input type="text"/>	<input type="text"/> to <input type="text"/>
		Filter Reset

A No significant similarity found. For reasons why, click here

4. Compared with RepC of *Agrobacterium tumefaciens*
< Edit Search Save Search Search Summary ▾

? How to read this report? ▶ BLAST Help Videos ↶ Back to Traditional Results Page

i Your search is limited to records that include: *Deinococcus radiodurans* R1 (taxid:243230)

Job Title	RepC (pTIR10, Agrobacterium tumefaciens)
RID	NPUDHUXH014 <small>Search expires on 09-12 20:12 pm</small> Download All ▾
Program	? Citation ▾
Database	nr See details ▾
Query ID	lcl Query_19892
Description	RepC (pTIR10, Agrobacterium tumefaciens)
Molecule type	amino acid
Query Length	404
Other reports	?

Filter Results

Percent Identity	E value	Query Coverage
<input type="text"/> to <input type="text"/>	<input type="text"/> to <input type="text"/>	<input type="text"/> to <input type="text"/>
		Filter Reset

A No significant similarity found. For reasons why, click here

5. Compared with RepC of *Shinorhizobium melliloti*

BLAST[®] » blastp suite » results for RID-NPU3TE2T014 Home Recent Results Saved Strategies Help

< Edit Search Save Search Search Summary
? How to read this report? BLAST Help Videos Back to Traditional Results Page

i Your search is limited to records that include: *Deinococcus radiodurans* R1 (taxid:243230)

Job Title	RepC:pSymA(Shinorhizobium meliloti)
RID	NPU3TE2T014 <small>Search expires on 09-12 20:07 pm</small> Download All ▼
Program	Citation ▼
Database	nr See details ▼
Query ID	lcl Query_58238
Description	RepC:pSymA(Shinorhizobium meliloti)
Molecule type	amino acid
Query Length	401
Other reports	?

Filter Results

Percent Identity	E value	Query Coverage
<input type="text"/> to <input type="text"/>	<input type="text"/> to <input type="text"/>	<input type="text"/> to <input type="text"/>
Filter		Reset

A No significant similarity found. For reasons why, click here

BLAST[®] » blastp suite » results for RID-SD5UHZPR016 Home Recent Results Saved Strategies Help

< Edit Search Save Search Search Summary
? How to read this report? BLAST Help Videos Back to Traditional Results Page

i Your search is limited to records that include: *Deinococcus radiodurans* (taxid:1299)

Job Title	RepN Bacillus subtilis
RID	SD5UHZPR016 <small>Search expires on 10-15 14:14 pm</small> Download All ▼
Program	Citation ▼
Database	nr See details ▼
Query ID	lcl Query_42053
Description	RepN Bacillus subtilis
Molecule type	amino acid
Query Length	287
Other reports	?

Filter Results

Percent Identity	E value	Query Coverage
<input type="text"/> to <input type="text"/>	<input type="text"/> to <input type="text"/>	<input type="text"/> to <input type="text"/>
Filter		Reset

A No significant similarity found. For reasons why, click here

BLAST[®] » blastp suite » results for RID-SD5HVB0016 Home Recent Results Saved Strategies Help

< Edit Search Save Search Search Summary
? How to read this report? BLAST Help Videos Back to Traditional Results Page

i Your search is limited to records that include: *Deinococcus radiodurans* (taxid:1299)

Job Title	RepN Staphylococcus aureus
RID	SD5HVB0016 <small>Search expires on 10-15 14:10 pm</small> Download All ▼
Program	Citation ▼
Database	nr See details ▼
Query ID	lcl Query_43517
Description	RepN Staphylococcus aureus
Molecule type	amino acid
Query Length	314
Other reports	?

Filter Results

Percent Identity	E value	Query Coverage
<input type="text"/> to <input type="text"/>	<input type="text"/> to <input type="text"/>	<input type="text"/> to <input type="text"/>
Filter		Reset

A No significant similarity found. For reasons why, click here

- a) The ability to delete the arrays is problematic if they represent the origin of replication, which would be essential for DNA replication of the affected chromosome, and so every and all daughter cells would lack this chromosome. Since this is not the case, the data indicate that these are accessory sequences rather than core sequences, and the latter are acted

upon by another protein. What are the putative initiators for Chr2 and MP? Are there homologues of other plasmid-like initiators, for example? The authors should discuss why it might be possible to delete these arrays in the first place.

b) *Author's response: Thank you. Please refer above response for most of the concern related to this query. Since, the genes encoding essential functions are located on chromosome I (primary chromosome) and majority of bacteria do not contain secondary chromosome, we were always questioning the essentiality of secondary genome elements in the normal growth of this bacterium. So, we attempted to answer the usefulness of secondary chromosome in this bacterium. We first ascertained that they are specific targets for DnaA and ParBs and then deleted these from the respective elements. Yes, we anticipated that all the mutant cells should be devoid of respective elements in cognate cis mutant but that did not happen, and some cells continue to show respective fluorescent foci. We failed in deleting putative ori of chromosome I and that was the most desirable control to interpret that the essentiality of secondary genome elements for normal growth. I agree with you that all the mutant cells were not devoid of respective secondary genome elements, which could be due to their maintenance by either an independent yet weak ori or canonical-ori independent support. The loss of these elements in respective cis mutant is a clear indication of a defect in their segregation. Some of these possibilities were discussed in original submission and made clearer in revision. Accordingly, conclusions have been moderated. Hope you agree.*

c) Are the entire arrays (all 11 copies) deleted in the chromosomal deletions?

Author's response: Yes, the entire arrays have been replaced with an expressing cassette of nptII in Chr II and megaplasmid.

c) Figs 7 and 8: how many cells have no copies of the affected chromosome? It is not possible to determine from the graphs (as drawn, it looks like no cells with 0 chromosomes). This number should be reported and is key to this discussion. Because the wild-type copy numbers of these chromosomes are 6-10 (from Fig 5A), it is possible that the chromosomes are essential as long as their copy number is 1 or greater. In other words, damaging replication or partition still allows cell growth as long as the chromosome is present, although lower copy number decreases gene dosage of important genes and affects growth rate/radiation resistance.

Author's response: Thank you so much. We have revisited the data and calculated the mutant's population with affected chromosome in ~200 cells and results are given in the revision. This analysis has produced interestingly sets of information but requires independent studies to speak about confidently. The wild type pattern for chromosome I has not affected in cisII and cisMP

deletion mutant supporting earlier conclusion that primary and secondary genome elements are maintained independently. The pattern of ChrII and MP is affected in terms of reduction in foci as well as many cells missing these replicons in respective mutants. Largely, it concurs the original conclusion that these elements house both replication and segregation functions. Furthermore, this bacterium exists in tetrad form where these cells share boundaries and the exchange of cellular components through membrane has not been ruled out for want of quality confocal imaging which would be done separately. Hope you appreciate that this is the first study which has generated materials with enough evidence to conclude some aspects and will allow us to study these things in more details independently. Some of these points have been clarified in revision.

[Figure removed by LSA Editorial Staff per authors' request]

- d) Fig 5B: How many copies of the chromosome are necessary to confer resistance to kanamycin? It is formally possible that a copy number of one would give Kan-sensitivity at the concentration of kanamycin used. In this case, the differences +/- kan could be explained as growth of cells with fewer copies rather than with no copies.

*Author's response:- Thank you for this insightful comment. We do not have single copy plasmid for *D. radiodurans* that would confer kanamycin resistance. So, at present we cannot answer it. However, when we create knockouts by replacement of target gene with *nptII* cassette, initially transformants having fewer copies integrated into genome have grown at this concentration of Kan. Subsequently, tolerance to Kan has increased in homogenous replacement where the number of copies would have become equivalent to number of copies of ploid genome elements per cell (6-10). From that experience, we believe that the concentration used (5µg/ml) for scoring CFU in this experiment can in principle be tolerated by single copy of *nptII*, ie if single copy of genome element carrying *nptII* was there in the cells. Since, we see all the four cells of tetrads are healthy in the presence of Kan and therefore, a possibility of communication from neighboring cells in tetrad cannot be ruled out. This suggest that *chrII* and *MP* seem to not require for the growth of this bacterium*

under normal conditions. We have modified our explanation as per your suggestions in the revision.

2. pg 17: The data do not allow the authors to conclude that there is a segregation role for the repeats. Either replication or segregation defects due to the deletions could account for chromosome and copy number loss, but I agree that the plasmid experiments support a replication role for the repeats. Mutations that damage ParB but not DnaA binding, or mutation of parB, for example, would be necessary to make specific conclusions about segregation. The statement that "the direct repeats function like ori and parS-like elements" should be clarified.

Author's response:- Thank you for comments. First, we checked the affinity of partitioning protein (ParB) with cognate cis elements in vitro. These cis elements have shown specific binding with their cognate as well as non-cognate ParBs. Furthermore, we have also found the affinity of cognate ParBs with decreasing number of repeats of corresponding cis elements albeit at varying levels. In FROS experiments, we have observed higher frequency of anucleate cells for affected replicons in respective cis mutants (e.g chrII in Δ cisII and Mp in Δ cisMP). Analysis of cell population of cisII and cisMP deletion mutant for the presence or absence of chrII and MP, respectively and its interpretation further supported both replication and segregation roles of these elements in this bacterium. There is no doubt that mutational studies as suggested would provide direct evidence of these elements' sites for ParB interaction. We strongly feel that the available evidence support parS role of these elements in this bacterium. Hope you agree.

3. pg 1: The TGS (tripartite genome segregation), or partition, system information is mis-stated and/or out of date. They are not necessarily "mostly" used, but they do contribute, typically in conjunction with SMC-like proteins and other systems. In addition, the push/pull models based on polymerization and depolymerization are not accepted for the ParABS chromosomal and plasmid systems; rather they refer to plasmid actin and tubulin-like partition systems. Although there is still debate as to exact details, the ParABS systems work differently.

Author's response: Thank you for the critical and valuable suggestions. I agree with you and accordingly introduction has been modified with this new information and appropriate references are cited in revision.

4. Figs 3 & 4: What is the "nsDNA" used in these experiments, and how is "molar" ratio determined?

Author's response: The non-specific DNA (nsDNA) was PCR amplified from genomic DNA of D. radiodurans using primers as shown in Table S1. The sizes of the product varied from 450bp to ~100bp and a nsDNA close to the size of target DNA was used in respective experiments.

Needless to mention that the approximate molar concentration was determined by considering the amount, molecular weight and volume in reaction mixture. Since the size of specific and non-specific target was nearly similar, the ratios were determined accordingly and were believed to be approximately close to that is given in manuscript. Hope we got your question correctly.

Reviewer #2 (Comments to the Authors (Required)):

Maurya and Misra extend this laboratory's previous characterization of genome replication in *Deinococcus radiodurans*. In this manuscript, they define trans and cis functions needed for the coordinate replication and segregation of Chromosome II and the mega-plasmid. They convincingly demonstrate that sequences upstream of the parAB loci on each genetic element (designated cisII and cisMP) are required bind to DnaA and their cognate ParB proteins and they provided inferential evidence that these elements are necessary for appropriate replication and segregation of the elements during growth. In general, the authors' conclusions are supported by the data provided. However, I have reservations concerning some of the authors' interpretations as indicated below.

Authors' response: Thank you so much for your kind words and appreciations of this work. Your critical comments have helped to further improve the manuscript. Hope you agree.

Comments

1. I am concerned by the following statement: "A majority of Δ cisII and Δ cisMP cells showed the loss of chrII and MP, respectively, while chr I localization pattern in these cells remained unperturbed when compared with wild type" and others like it in the manuscript. It seems that the authors are suggesting that chrII and the MP are dispensable and that the species does not require any of the genes for survival. Is this what is being implied? The authors thoughts on this are not clear. Also, it is not obvious why the authors believe a majority (and not all) Δ cisII and Δ cisMP cells show loss of chrII and MP. Given the role and importance they ascribe to these sequences, it would seem all cells would be affected.

Author's response: Thank you for comments. Reviewer 1 also has nearly similar concerns and that has been addressed above at our best. In this study, we have checked the survival of both Δ cisII and Δ cisMP cells under normal and radiation stressed conditions and found that these mutants are extremely sensitive to ionizing radiation and show poor survival than the growth obtained under normal conditions. These observations suggest that secondary genome replicons are responsible for survival under radiation stressed condition rather than normal. In recent review article of 2017 (PMID: 28794225), diCenzo and Finan have reported that "secondary replicons in bacterial genomes carry no core genes and are nonessential and thus dispensable for cell viability in most environments". In FROS study, we found that majority cells in each

tetrad of the cis mutant have lost the affected replicon. This would have occurred due to defective segregation in the absence of cognate cis elements. The data shown above also indicated that the number of cells containing chrII and MP has reduced in cognate cis mutant as compared to wild type. While detailed studies would be needed for precise answer of some of these obvious questions, the available data support the role of these elements in both replication and segregation. The text has been moderated in revision. Hope you agree.

2. I am unsure about the authors' description and explanation for the ionizing radiation sensitivity of the Δ cisII and Δ cisMP cells. The authors state, "Since, the deletion of these cis elements had caused reduction in the copy number of cognate replicon and affected the distribution of the secondary replicons in daughter cells, the loss of gamma radiation resistance could be implicated to either reduction in copy number or complete loss of secondary genome replicons in these mutants." This statement indicates, rightly so, that loss of these replicons results in sensitivity. It seems this would be the consequence of the loss of know gene functions associated with chrII and MP. This possibility is not discussed extensively, leading me to wonder if the authors are implying another more obtuse mechanism responsible of loss of radioresistance.

Author's response: Thank you for your comment and an admirable curiosity. True, our results indicated that secondary genome elements are mostly involved in increasing the fitness of bacterium under stress conditions. A review covering the genome analysis of nearly 60 bacteria having multipartite genome system and ploidy has argued that most of them must increase their fitness either under biotic stress in host or abiotic stress in environments. This review is cited in this paper. So there seems to be a strong growing evidence to suggest that secondary genome elements contribute in stress resistance in bacteria. This has been speculated in this bacterium also but there was no detailed studies to support it. Through this work, we provide evidence that the genome copy number and presence /absence of secondary genome elements make major difference in radioresistance of this bacterium. In response to your comments, we have discussed the genes present on these genome elements and their roles in radioresistance in the revision. Hope you agree.

3. Finally, I am struck by the following statement. "Survival of other three cells in tetrad without replicons conferring antibiotic resistance under selection pressure is intriguing and a strong possibility of cross protection from the cell conferring antibiotic resistance cannot be ruled out." This suggests a fascinating possibility - one not mentioned by the authors - that there is an intercommunication between cells carrying the missing replicons and those that are not. I should like the authors to consider the fact that there is a period in which segregating cells are in communication. Long-lived proteins made in a cell with the replicons could act in a cell without the replicons, passing through the septal annulus as the cells divide.

Author's response : *Thank you for this comment. We did not have any evidence to explain why 3 of the 4 cells in tetrad do not show replicon that vehicles antibiotic resistance and FROS are still surviving under pressure. There can be many explanations, the most acceptable one is believed to be through intercommunications between the cells. The possibility of exchange of cellular materials between the cells in tetrad would be worth investigating in detail. Dr Abraham Minsky's group have published high resolution TEM pictures of Deinococcus radiodurans and have shown that there is a communication between the cells in tetrad (Science 299: 254-256). All these are circumstantial evidence and further details have started coming now. We are trying to understand the role of eDNA in radioresistance. I appreciate your questions and would be worth looking forward for providing some answers independently. We have modified these sentences in revision. Hope you agree.*

We thank both the reviewers for their critical comments and editor for getting this work nicely evaluated by experts in the field. We appreciate the most constructive comments and optimistic suggestions by both the reviewers, which have helped us to further improve this manuscript. We are very much hopeful that our decision to publish this work in LSA is judicious that will give me an opportunity and special reason to enjoy its success. Hope you all find revision acceptable for publication in LSA.

H S Misra

(For co-authors)

November 2, 2020

RE: Life Science Alliance Manuscript #LSA-2020-00856-TR

Author information redacted

Dear Dr. Misra,

Thank you for submitting your revised manuscript entitled "Studies on ori and parS-like functions in secondary genome replicons in *Deinococcus radiodurans*". We would be happy to publish your paper in Life Science Alliance pending final revisions necessary to meet our formatting guidelines.

Along with the points listed below, please also attend to the following:

- please add a callout for Figure 1 A,B; Figure 3 A-H; Figure 6 A-C; Figure 7 A-C; Figure 8 A-C; Figure S2 A-C; Figure S3 A,B; Figure S4B; Figure S5A,B to your main manuscript text
- Please re-check the figure panels and associated callouts and figure legends for Fig 9 and Fig 5D
 - Fig 9 has panels A-C called out but there are not such panels in the figure. There are legends for Fig 9 A,B,C but these are missing from the figure.
 - There is a legend for Fig 5D but it's missing from the figure
- The 3 schematics shown in Figure S3 use the same image for *D. radiodurans*. While we understand that this is not a data image, would it be possible to include separate images for each schematic? If you think it is necessary to keep the image same (for showing consistency in the editing strategy), we request you to clarify this in the legend.

A. FINAL FILES:

-- Summary blurb (enter in submission system): A short text summarizing in a single sentence the study (max. 200 characters including spaces). This text is used in conjunction with the titles of papers, hence should be informative and complementary to the title. It should describe the context

and significance of the findings for a general readership; it should be written in the present tense and refer to the work in the third person. Author names should not be mentioned.

B. MANUSCRIPT ORGANIZATION AND FORMATTING:

Sincerely,

Shachi Bhatt, Ph.D.
Executive Editor
Life Science Alliance
<https://www.lsjournal.org/>
Tweet @SciBhatt @LSAJournal

November 4, 2020

RE: Life Science Alliance Manuscript #LSA-2020-00856-TRR

Author information redacted

Dear Dr. Misra,

Thank you for submitting your Research Article entitled "Studies on ori and parS-like functions in secondary genome replicons in *Deinococcus radiodurans*". It is a pleasure to let you know that your manuscript is now accepted for publication in Life Science Alliance. Congratulations on this interesting work.

IMPORTANT: The manuscript text is still missing a callout for S5A,B. Please add that in during the proofs stage.

DISTRIBUTION OF MATERIALS:

Again, congratulations on a very nice paper. I hope you found the review process to be constructive and are pleased with how the manuscript was handled editorially. We look forward to future exciting submissions from your lab.

Sincerely,

Shachi Bhatt, Ph.D.

Executive Editor

Life Science Alliance

<https://www.lsjournal.org/>
